# VeriDark: A Large-Scale Benchmark for Authorship Verification on the Dark Web

**Andrei Manolache**\*
Bitdefender
amanolache@bitdefender.com

**Florin Brad**\*
Bitdefender
fbrad@bitdefender.com

**Antonio Barbalau**
University of Bucharest
abarbalau@fmi.unibuc.ro

**Radu Tudor Ionescu**
University of Bucharest
raducu.ionescu@gmail.com

**Marius Popescu**
University of Bucharest
popescunmarius@gmail.com

## Abstract

The *Dark Web* represents a hotbed for illicit activity, where users communicate on different market forums in order to exchange goods and services. Law enforcement agencies benefit from forensic tools that perform authorship analysis, in order to identify and profile users based on their textual content. However, authorship analysis has been traditionally studied using corpora featuring literary texts such as fragments from novels or fan fiction, which may not be suitable in a cybercrime context. Moreover, the few works that employ authorship analysis tools for cybercrime prevention usually employ ad-hoc experimental setups and datasets. To address these issues, we release `VeriDark`: a benchmark comprised of three large scale *authorship verification* datasets and one *authorship identification* dataset obtained from user activity from either *Dark Web* related Reddit communities or popular illicit *Dark Web* market forums. We evaluate competitive NLP baselines on the three datasets and perform an analysis of the predictions to better understand the limitations of such approaches. We make the datasets and baselines publicly available at `https://github.com/bit-ml/VeriDark`.

## 1 Introduction

The Dark Web is content found on the DarkNet, which is a restricted network of computers accessible using special software and communication protocols, such as Tor[2]. This infrastructure offers anonymity and protection from surveillance and tracking, which greatly benefits users whose privacy is of great concern (*e.g.* journalists, whistleblowers, or people under oppressive regimes). However, cybercriminals often exploit these services to conduct illegal activities via discussion forums and illicit shops corresponding to different marketplaces, frequently referred to as *hidden services*.

Law enforcement agencies have started to crack down on DarkNet cybercriminals in the last decade. One of the largest marketplaces was Silk Road, estimated to host between 30000 and 150000 active customers [Christin, 2013], which was shutdown by the FBI in 2013. However, this success was short-lived due to the quick shift of the customer base and vendors to other marketplaces. As of

---

\*Equal contribution.
[2] `https://www.torproject.org/`

**Agora**

A reporter incapable of accessing **DNMs** without lots of help is **[rpbably** incapable of giving a balanced report. This is exactly the reason i tried **:)** i was pretty scared from what he would write without help. As far as know **they are not scammers at all**, just low on activity and been working mostly with the German **market** since **SR1** days as **"Drugs&Bets"** that was later changed to Outlaw - Administration changed along the way and former admins opened Area51 **market**. At some point a post on a German forum posted PM's from the **market** and claimed he **hacked** the **market** - the admin claimed it was from a **hacked** account (the real reason remains unclear since **PGP** is enforced on the **market**) The retired competitors from Area51 **Attacked** the **market** and caused it to be down for a while. But they fixed all issues quickly and implemented other **security features**. No money was lost. and no "we got **hacked** and will repay" claims were made **:)**

**PAN**

**Lutta**. Was this a work of **nachtmagen**? There was that possibility, faded as the yellow craft was. Then who had performed it? **Coryn** could not grab it; he snatched at empty air. There was nothing to tell him. Nothing that anything could tell him, nor aid him. This was an enigma beyond most anagrams of matters. This was a mess. He had found her, **Lutta**, merely lying on some branch of the tree. And, through some urging of the gizzard, he had brought her back into the tree itself. This made little sense. He dived into another pool of thoughts. What if...? There was a bang; he jumped and turned. There was a curse of **"raccdrops**!" Twilight. **Coryn** called, one wing still laid flat over a book, **"What's going on?"** More cursing, followed by **Gylfie**'s reprimands. There was a loud whirlwind of a flutter, and soon the whole Band was within the room! **"What?"** **Coryn** said loudly at the undignified relative of a heap before him. **"What's happened?"**

Figure 1: An Agora Dark Web forum post compared to a PAN 2020 authorship verification dataset sample. Notice the different themes (*drugs*, *hacking* vs. *fiction*) and style (*colloquial*, *sloppy* vs. *narration*, *dialogue*). The Dark Web posts often include emojis and acronyms of DarkNet-related concepts, while PAN samples contain fictional character names and fabricated words.

2022, relocation to other marketplaces remains a key issue when dealing with DarkNet cybercrime [Europol, 2021]. This indicates that more efforts are needed to develop law enforcement forensic tools to help analysts. Since one of the largest digital footprints of a Dark Web user is their post history, developing text-based authorship analysis software enables these efforts.

The authorship analysis field has a long history. The first efforts go back to as early as the nineteenth century [Mendenhall, 1887], followed by the seminal work of Mosteller and Wallace [1964] during the mid-twentieth century, where the authors used Bayesian statistical methods to identify the authors of the controversial *Federalist* papers. The domain of authorship analysis spans a range of different tasks, such as: authorship attribution, which is the task of identifying the author of a text given a *closed set* of identities (*i.e.* a list of known authors); authorship verification (AV), the task of determining if two texts A and B are written by the same author or not; author profiling, where certain characteristics (*e.g.* age, gender, nationality) need to be predicted.

Research in the past decade has been spearheaded by machine learning approaches coupled with increasingly larger corpora and more diverse corpora [Brocardo et al., 2013]. However, these efforts were mainly targeting literary works in the form of novels and fan fiction [Kestemont et al., 2020], with few works targeting large-scale noisy data collected from forums and discussion platforms [Zhu and Jurgens, 2021, Wegmann et al., 2022]. As such, authorship verification solutions developed using these datasets are likely to underperform under a domain shift when deployed on Dark Web corpora, which have a different linguistic style and register (domain-specific abbreviations and code names, slang, a particular netiquette, presence of emojis, etc.), as exemplified in Fig. 1.

We thus introduce the `VeriDark` benchmark, which contains three large-scale *authorship verification* datasets collected from social media. The first one is `DarkReddit+`, a dataset collected from a Reddit forum called `/r/darknetmarkets`, which features discussions related to trading on the DarkNet marketplaces. The next two datasets, `Agora` and `SilkRoad1`, are collected from two Dark Web forums associated with the two most popular defunct marketplaces (Agora and Silk Road). Our aim is twofold: *(i)* to enable the development and evaluation of modern AV models in a cybersecurity forensics context; *(ii)* to facilitate the evaluation of existing AV methods under a domain shift.

Moreover, we introduce a small authorship identification task in the cybersecurity context. Specifically, given a comment, the task asks to predict the user that made the comment. The task is a 10-way classification problem, where the classes are given by the top 10 most active users. The train set contains comments from the subreddit `/r/darknetmarkets` only, but we test on both `/r/darknetmarkets` and other subreddits unrelated to DarkNet discussions.

Table 1: Statistics for our proposed datasets (**bold**) versus other relevant datasets. **Agora** and **SilkRoad1** are the largest authorship verification datasets to date in terms of the number of examples, while having much shorter texts on average than the PAN datasets. To our knowledge, **Agora** and **SilkRoad1** are the first publicly available authorship verification datasets featuring texts from the Dark Web. **DarkReddit+** is used for both the authorship verification and identification tasks. The row **Total** is obtained by merging **Agora**, **SilkRoad1** and **DarkReddit+**. The DarkReddit [2021] line refers to a much smaller collection of data which was mainly utilized for evaluating the few-shot capabilities of models trained on the PAN corpora.

| Dataset | Train | Valid | Test | #auth. | Avg #words | Source | Task |
|---|---|---|---|---|---|---|---|
| **Agora** | 4,195,381 | 216,570 | 219,171 | 12,159 | 143 | Dark Web | AV |
| **SilkRoad1** | 614,656 | 34,300 | 32,255 | 30,206 | 119 | Dark Web | AV |
| **DarkReddit+** | 106,252 | 6,124 | 6,633 | 17,879 | 84 | Clear Web | AV |
| **Total** | 4,916,289 | 256,994 | 258,059 | 60,244 | 135 | Mixed | AV |
| **DarkReddit+** | 6,817 | 2,275 | 2,276 | 10 | 107 | Clear Web | AI |
| DarkReddit [2021] | 204 | 412 | 412 | 117 | 524 | Clear Web | AV |
| PAN20-small [2020] | 52,601 | - | - | 52,655 | 3,920 | Clear Web | AV |
| PAN20-large [2020] | 275,565 | - | - | 278,169 | 3,920 | Clear Web | AV |

We make the datasets publicly available[3] and we also maintain a public leaderboard[4].

## 2  Related Work

Our work relates to DarkNet forensic investigations and author profiling. One of the first works exploring the Dark Web hidden services is an analysis of the Silk Road 1 marketplace by Christin [2013], where the author examined and described the inner-workings of the defunct illegal marketplace. Undertaking a similar challenge, Ursani et al. [2021] used an anomaly detection method to study how the DarkNet markets react to certain disruptive events, such as Europol's Operation Onymous [Europol, 2014]. In addition to studying the Dark Web, researchers endeavored to investigate drug trafficking on the clear web. Mackey et al. [2017], Mackey and Kalyanam [2017], Li et al. [2019], Qian et al. [2021] have employed various machine learning models to identify illegal drug postings on Instagram and Twitter. To support this kind of pursuits, Bogensperger et al. [2021] crowdsourced a dataset for Drug Named Entity Recognition in the DarkNet marketplaces.

Dark Web author profiling historically employed various techniques and multimodal data. In both [Wang et al., 2018] and [Jeziorowski et al., 2020], the authors linked multiple accounts (*Sybils*) of vendors using photos of their products. Similarly, Zhang et al. [2019] proposed a model that leverages both images and descriptions of drug listings to create a low-dimensional *vendor embedding* which can be used to determine *Sybils* across different marketplaces. In Maneriker et al. [2021], the authors proved that multitask learning can be successfully employed for authorship attribution across DarkNet markets, while Arabnezhad et al. [2020] showed that it is possible to link the activity of users in the illegal forums with their activity on Reddit using stylometric approaches for authorship attribution. Although both previously mentioned papers have similar goals to our work, we strive to tackle the more general authorship verification task, whereas their goal is to employ authorship attribution for the Dark Web. Moreover, our main focus is providing an unified benchmark for authorship tasks on noisy Dark Web data, as opposed to solely model-related improvements.

More broadly, our work relates to authorship attribution and authorship verification, which is a more recent and difficult task. Historically, machine learning models have been successfully deployed for the authorship attribution problem on small datasets, some of the more prevalent techniques employing decision trees [Zhao and Zobel, 2005] and Support Vector Machines [de Vel et al., 2001]. Machine learning methods have also found success in the authorship verification task, where the unmasking technique has been particularly influential [Koppel et al., 2007, Bevendorff et al., 2019b]. Recently, the direction has shifted towards using large scale datasets and neural networks [Boenninghoff et al., 2019, 2020, 2021, Ordoñez et al., 2020, Soto et al., 2021, Tyo et al., 2021].

---

[3] https://github.com/bit-ml/VeriDark
[4] https://veridark.github.io/

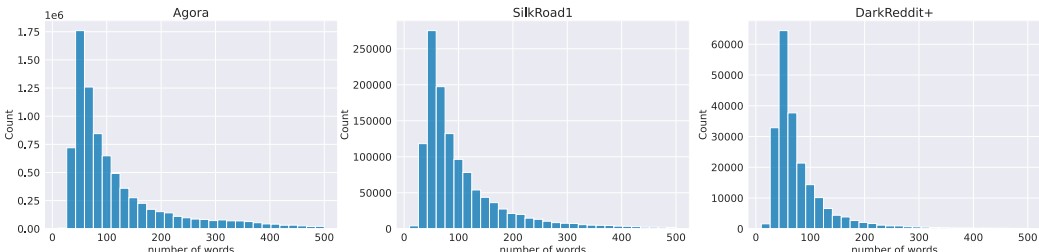

Figure 2: Histogram of text lengths (as number of words) for all the three datasets.

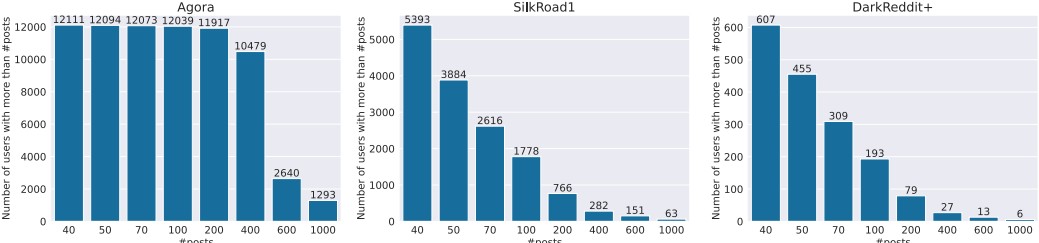

Figure 3: Bar plot of the users' activity at different post counts. The users on `DarkReddit+` and `SilkRoad1` have similar number of posts distributions, while `Agora` has a large pool of active users.

These advancements primarily result from the efforts of the CLEF Initiative, which released the first large-scale authorship verification datasets and tasks [Kestemont et al., 2019, 2020, 2021].

Distinctly from the PAN authorship verification datasets, which feature literary texts, we tackle authorship verification in the cybersecurity context, where there is a significant domain shift towards a more informal and concise writing style, as can be seen in Fig. 1.

## 3 VeriDark Authorship Verification Datasets

For all the authorship verification datasets, we removed the PGP signatures, keys and messages, since they can contain information which could leak the user's identity. We then filtered the duplicate messages and removed the non-English messages using the *langdetect*[5] library. We removed comments with less than 200 characters due to very little information available to decide the authorship status. This preprocessing step reduced the `Agora` and `SilkRoad1` dataset sizes by a factor of 2, and `DarkReddit+` by a factor of 4. Then, we partition the list of authors into three disjoint sets: 90% of the authors are kept for the train set, 5% of the authors are kept for the validation set, and another 5% are kept for the test set. We thus create a more difficult *open-set* authorship verification setup, in which documents at test time belong to authors that haven't been seen during training time. The training, validation and test split ratios remain approximately 90%, 5% and 5%, respectively.

### 3.1 DarkReddit+ Authorship Verification Dataset

We build a large AV dataset called `DarkReddit+`, which contains same author (SA) and different author (DA) pairs of comments from the `/r/darknetmarkets` subreddit. This defunct subreddit was a gateway to the DarkNet and hosted discussions about illicit services and goods. These comments were written between January 2014 and May 2015. The data was retrieved from a large archive[6] containing all the comments written on `reddit.com` up until 2015 [Baumgartner et al., 2020].

We generate same author pairs by iterating through each author, then selecting document pairs until all its documents are exhausted. We then generate DA pairs using the following procedure: $(i)$ select two random authors $a_1$ and $a_2$, $(ii)$ select a random document from author $a_1$ and a random document from author $a_2$, $(iii)$ repeat the first two steps until we obtain the same number of DA and SA pairs,

---

[5]https://github.com/fedelopez77/langdetect
[6]https://reddit.com/r/datasets/comments/3bxlg7/i_have_every_publicly_available_reddit_comment/

Table 2: Results on the `VeriDark` authorship datasets: `DarkReddit+` (**DR+**), `SilkRoad1` (**SR1**), `Agora` (**AG**) and All (the previous three datasets aggregated). The best results for each fixed train set are colored in brown, while the second best results are colored in blue. Unsurprisingly, the best average test metrics are obtained when training on the same dataset. Notice the good cross-dataset performance transfer when training on one dataset and testing on the other two. Training on All datasets performs slightly worse than training on the same dataset as the test set, but outperforms the cross-dataset strategy, showing that more data can make a model more robust across datasets. *The mean over 5 runs is reported, with the *std* line being the standard deviation for the *avg.* score.

| Train | DarkReddit+ | | | | SilkRoad 1 | | | | Agora | | | | All | | | |
|---|---|---|---|---|---|---|---|---|---|---|---|---|---|---|---|---|
| **Test** | **DR+** | **SR1** | **AG** | **All** | **DR+** | **SR1** | **AG** | **All** | **DR+** | **SR1** | **AG** | **All** | **DR+** | **SR1** | **AG** | **All** |
| *F1* | 75.0 | 70.2 | 72.7 | 72.4 | 75.3 | 81.6 | 80.2 | 80.3 | 73.4 | 79.0 | 84.9 | 83.8 | 75.3 | 81.6 | 85.8 | 84.9 |
| *F0.5* | 75.1 | 76.8 | 78.9 | 78.5 | 69.7 | 83.0 | 79.7 | 79.8 | 66.1 | 76.3 | 85.4 | 83.6 | 70.1 | 80.6 | 86.2 | 84.9 |
| *c@1* | 75.1 | 74.8 | 76.2 | 76.0 | 71.5 | 82.6 | 80.2 | 80.3 | 67.3 | 78.3 | 85.3 | 83.9 | 71.7 | 81.7 | 86.1 | 85.2 |
| *ROC* | 83.6 | 85.1 | 86.1 | 85.9 | 81.8 | 91.0 | 88.3 | 88.4 | 80.3 | 87.1 | 93.5 | 92.3 | 82.0 | 90.2 | 94.2 | 93.5 |
| *avg.* | 77.2 | 76.7 | 78.4 | 78.2 | 74.6 | 84.6 | 82.1 | 82.2 | 71.8 | 80.2 | 87.3 | 85.9 | 74.7 | 83.5 | 88.1 | 87.1 |
| *std** | 0.27 | 1.14 | 1.75 | 1.64 | 0.67 | 0.23 | 0.33 | 0.28 | 0.72 | 0.55 | 0.39 | 0.34 | 1.08 | 0.54 | 0.16 | 0.19 |

resulting in balanced classes. We obtain a total number of almost 120k pairs, which are divided into approximately 107k training pairs, 6k validation pairs and 6k test pairs, as shown in Table 1.

## 3.2 DarkNet Authorship Verification Datasets

Table 3: Performance on the `VeriDark` datasets when training on the PAN dataset. The best results are colored in brown, while the second best results are colored in blue. Notice the lower performance compared to the models trained on the `VeriDark` datasets, shown in Table 2.

| Train | PAN2020 large | | | |
|---|---|---|---|---|
| **Test** | **SR1** | **DR+** | **All** | **AG** |
| *F1* | 59.7 | 61.2 | 62.5 | 63.0 |
| *F0.5* | 67.2 | 67.2 | 68.3 | 68.5 |
| *c@1* | 66.2 | 66.2 | 67.5 | 67.7 |
| *ROC* | 72.8 | 72.8 | 73.7 | 73.6 |
| *avg.* | 66.1 | 66.9 | 68.0 | 68.2 |

The next two datasets, called `SilkRoad1` and `Agora`, were obtained from comments retrieved from two of the most popular DarkNet marketplace forums: Silk Road and Agora. Silk Road was one of the largest black markets, which started in 2011 and was shut down in 2013. We used publicly available forums scraped from DarkNet from 2013 to 2015 [Branwen, 2021].

The two archived forums have different folder structures, due to crawling differences, but share the same SimpleMachine-Forums 2 forum layout [7]. To extract the comments, we first retrieved all the HTML files that contained the word 'topic' for both forums. We then parsed each HTML file using the BeautifulSoup library [8], extracting the author names and the comments and storing them in a dictionary. We generated same author and different author pairs using the same procedure used for the `DarkReddit+` dataset. Statistics for both datasets are listed in Table 1. The `Agora` dataset is almost 7 times larger than the `SilkRoad1` dataset, which in turn is almost 7 times larger than the `DarkReddit+` dataset. To our knowledge, the DarkNet datasets are the largest authorship verification datasets to date in terms of the number of examples. However, distinctly from the PAN text pairs, which feature large fanfiction excerpts, the DarkNet examples are much shorter (average number of words is around 120), due to the nature of the short and frequent forum interactions. We plot the histograms of the text lengths for all three datasets, where the length of a text is given by the number of words. All our datasets have similar length distributions, as illustrated in Figure 2, but `DarkReddit+` has shorter comments overall, with very few comments over 200 words. When it comes to user activity, users in `SilkRoad1` and `DarkReddit+` have similar activity distributions, with a large number of users with low activity, while `Agora` users are more active, as can be seen in Figure 3.

---

[7] https://www.simplemachines.org
[8] https://www.crummy.com/software/BeautifulSoup/bs4/doc/

$$P(\textit{same}|A, B) = \frac{1}{k} \sum_{i=1}^{k} P(\textit{same}|c_i\,c_i)$$

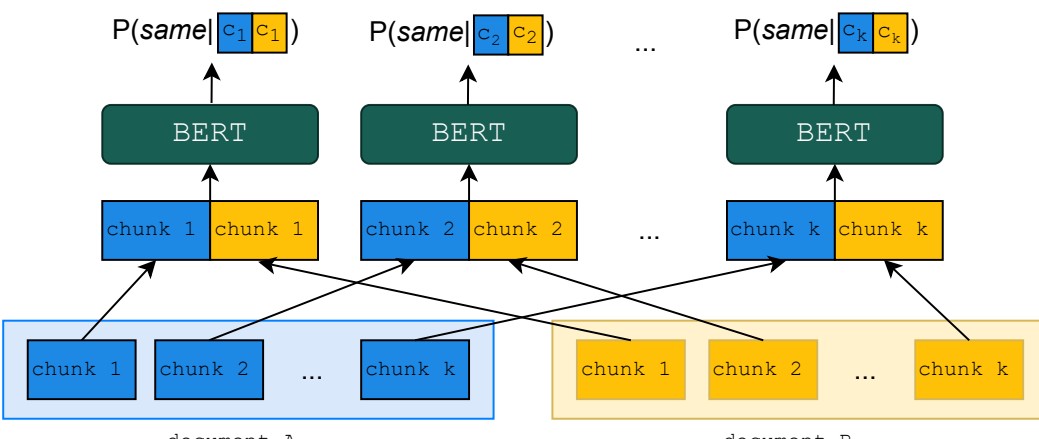

Figure 4: Authorship decision based on the aggregated probability of the same author over chunk pairs from both documents A and B. Each document is split into fixed-length chunks. The longer document is trimmed such that both documents contain $k$ chunks each.

### 3.3   DarkReddit+ Author Identification Dataset

A forensic scenario of interest is identifying users on the Web based on their activity on the Dark Web. To implement this scenario, we could leverage our pre-trained authorship verification models to perform authorship attribution by successively comparing our target text with the candidate texts belonging to our pool of selected authors. However, in our particular setup, this would require at least ten separate passes through the authorship verification model (assuming that both documents contain at most 256 tokens).

An easier and more efficient task would be to identify a Dark Web user based solely on their comment. To this end, we employ a three-step preprocessing procedure. We first retrieve all the comments that users from /r/darknetmarkets have written in other subreddits as well. We refer to Reddit comments which are not Dark Web related as ClearReddit. Second, we keep the users that have at least 5 comments in both DarkReddit+ and ClearReddit. Finally, we select the top 10 most active users in DarkReddit+, based on the number of comments. Our motivation for choosing 10 authors for the identification problem is two-fold: i) this is the number of authors used by the PAN authorship attribution task[9]; ii) only a small number of authors were active in both DarkReddit+ and ClearReddit, therefore a higher number of authors would have resulted in a very imbalanced dataset with respect to the number of posts per author. We thus create a classification task in which, given a comment, the correct user (out of 10 possible users) must be predicted. We list the train, validation and test splits in Table 1.

## 4   Experiments

Next, we describe our training methodology, models, and general experimental setup. For all our experiments, we use the train/validation/test splits as described in Sec. 3.

### 4.1   Authorship Verification

**Training.** We fine-tune BERT-based models [Devlin et al., 2019] on our dataset as binary classifiers, as suggested in Manolache et al. [2021]. During training, given two texts $S_1$ and $S_2$, we randomly

---

[9] https://pan.webis.de/clef19/pan19-web/authorship-attribution.html

select 254 tokens from $S_1$ and 254 tokens from $S_2$. We concatenate the two token sequences, separating them using the $[SEP]$ special token. If the texts are too short, we append the $[PAD]$ special token to the sequence, until we obtain a sequence of length 512. The tokens are then forwarded to the backbone and we obtain a joint sequence pair embedding $h_{[CLS]}$. Finally, we feed the obtained embedding to a linear layer and optimize the whole network using the binary cross-entropy loss.

**Evaluation.** We evaluate the model as follows: given two texts $S_1$ and $S_2$, we split each document into chunks of $254$ tokens. We trim the document with more chunks, such that both documents have the same number of $K$ chunks. We then make $K$ pairs of chunks, by picking the chunks with the same indices, as can be seen in Figure 4. We follow the training procedure and construct the input sequence by separating the two chunks with [SEP] tokens and prepending them with the [CLS] token. Finally, we separately feed the $K$ pairs into the model and obtain the probabilities for the two classes. We average the probabilities for the final prediction. For a better understanding of the models' performance, we evaluate them using the same metrics as in the PAN authorship verification shared tasks [Kestemont et al., 2019], namely the $F1$, $F0.5$, $C@1$ and $AUROC$ scores, with $avg.$ being the average over the previously mentioned metrics.

We look at in-domain vs. cross-domain performance for each of the VeriDark datasets in Table 2. The best performance on each test dataset is obtained when training on the same dataset, with the exception of Agora, where the results are slightly worse than training on All datasets. Cross-domain performance levels when training on the VeriDark datasets are significantly better than the cross-domain results when training on the PAN dataset, as shown in Table 3. This suggests that, in the forensics context, it is important to train authorship methods on data related to cybersecurity.

We trained our models on RTX Titan and RTX 2080 GPU cards on an internal cluster. The estimated time to reproduce all the results is four days.

## 4.2 Author Identification

Results on the author identification task are featured in Table 4. We use five training setups and four testing scenarios. First, we simply train and evaluate our model on the proposed DarkReddit+ author identification dataset. Then, in order to simulate a more realistic scenario, in which a post does not belong to any of the investigated authors, we introduce a new '*Other*' class. For this additional class, we select a number of samples equal to the average number of samples per author. These are randomly chosen from a secondary source: either the VeriDark AV corpora or the PAN dataset. In Table 4, each column name denotes the source of the samples comprising the '*Other*' class during training. Results from the column 'None' are from a model trained on the ten authors only, without posts from the 'Other' class. We fine-tune five BERT-based models, one for each of the corresponding training source. The 'None' column corresponds to a 10-way classifier (10 authors), while the other columns correspond to 11-way classifiers (10 authors plus 'Other' training examples). Models are tested on texts from both DarkReddit+ (first two rows) and ClearReddit (last two rows). The former test setup refers to identifying users from their DarkReddit+ comments, based on knowledge of their activity on DarkReddit+, while the latter test setup refers to identifying users from their ClearReddit comments, based on the same activity. For both test domains, we evaluate the models with test examples from the ten authors only ('W/o Other' rows). We also evaluate the models with test examples from the ten authors plus additional test examples from the Other class, corresponding to the Secondary Source domain ('With Other' rows). The performance difference between the 'W/o Other' and 'With Other' scenarios comes solely from the extra data used during evaluation.

Experiments show that the performance increases only when adding training data from PAN and testing on the DarkReddit+ examples. In this case, the model learns to better distinguish the authors by learning an additional 'Other' class whose examples are significantly different from the training set (PAN vs DarkReddit+). However, when the domain of the 'Other' class is similar or the same as the training set (SilkRoad, Agora, DarkReddit+), the model gets worse at recognizing the authors based on their DarkReddit+ comments. Moreover, this observation holds when testing on authors considering their ClearReddit comments, regardless of the domain of the 'Other' class.

To conclude, we only observe performance gains when introducing an additional 'Other' class whose domain is significantly different from the training set as well as the test set. We note that our baselines are competitive, while also allowing a lot of room for improvements. As further work, it would be

Table 4: Results on the `DarkReddit+` Author Identification Dataset in terms of accuracy. Each column name refers to the source of the '*Other*' training samples. The model from the **None** column has been trained on texts from the ten authors only. The first two rows show the results of the models evaluated on the `DarkReddit+` test set, while the last two rows show results on the `ClearReddit` test set. The first row of each section presents results on classifying texts belonging only to the ten original authors. For the second row of each section, texts from the '*Other*' class were introduced at test time from the secondary source.

| | | | Secondary Source | | | | |
|---|---|---|---|---|---|---|---|
| | | | None | PAN | SilkRoad 1 | Agora | DarkReddit+ AV |
| Test Setup | Dark | W/o *Other* | 84.6 | 85.2 | 83.9 | 83.3 | 82.0 |
| | | With *Other* | - | 86.5 | 83.7 | 83.5 | 80.3 |
| | Clear | W/o *Other* | 81.2 | 78.4 | 76.5 | 76.8 | 74.8 |
| | | With *Other* | - | 80.4 | 75.0 | 77.7 | 73.9 |

interesting to alleviate the performance penalty for the situations where the 'Other' class distribution is more similar to `DarkReddit+`.

## 4.3 Error Analysis

We perform a qualitative analysis by inspecting both true positive (same author correctly detected) and false positive (different author pairs predicted as having the same author) predictions. We notice that the first two examples shown in Table 5 are wrongly classified as having the same author, which may be due to similar punctuation (multiple '*!*'), same abbreviations ('*u*'), same writing style (negative feedback) or multiple occurrences of some named entities (drug names). The last two same author pairs are correctly classified, which may be due to many shared named entities in the fourth example (vendor name, drug name, country) or similar computer science jargon (*encrypted*, *whonix*).

The shared named entities, while being suggestive of same authors, may represent spurious features, which the model will likely exploit, leading to false positives. The importance of mitigating the effect of named entities has also been noted by Bevendorff et al. [2019a] and Manolache et al. [2021]. Since there are many types of named entities that can be removed (vendor names, drug names, usernames, places, etc.), we do not perform any such deletion and instead leave this preprocessing decision to be made during model creation.

## 4.4 Limitations and Further Work

**Noisy negative examples.** One of the underlying assumptions when collecting texts for authorship analysis from different discussion and publishing platforms is that that distinct authors have distinct usernames. However, this may sometimes not be the case, as people often have multiple accounts and, as a result, write under several pseudonyms or usernames. These accounts are known as *Sybils* in the cybersecurity context and belong to either DarkNet vendors or users with multiple identities. As a result of this phenomenon, some different author pairs may actually have the wrong label, since they are written by the same author. This may hurt the training process, which may distance same author document representations instead of making them more similar. The evaluation may also suffer due to a model that may correctly predict *same author* but will be penalized due to the wrong label *different author*.

**Noisy positive examples.** Another related phenomenon may arise from multiple users writing under the same account. This leads to examples being labeled as *same author* when in fact the correct label is *different author*. Such examples may again affect the learning process, which tries to draw closer different author representations instead of distancing them. The evaluation performance may decrease from correctly classifying *different author* pairs, which are mislabeled as being *same author*. Both phenomena are not inherent to our proposed datasets and may also arise in other authorship verification benchmarks, such as PAN, where an author may write fanfictions under multiple pseudonyms, or multiple authors may share the same account. However, we acknowledge

Table 5: Qualitative analysis for several pairs from the `Agora` test set. Some sensitive words (such as drug names or vendor names) have been censored. The first two examples are written by different authors, but predicted as having the same author. The first prediction may be triggered by similar punctuation (multiple exclamation marks). The second and third same-author pairs share many named entities (drug name, vendor, country), but the second one is misclassified. The fourth same-author example may be correctly classified due to similar specialized vocabularies (*encrypted*, *whonix*).

| # | Pred. | Text A | Text B |
|---|---|---|---|
| 1 | FP | *(...)* wouldnt be like *[DRUG-A]* at all**!!!** *[DRUG-B]* overpowers any sensation that *[DRUG-A]* gives**!!!** sounds like **u** got jipped**!!!!** *[DRUG-A]* that doesent numb for me is unheard of**!!!** *(...)* | none at all friend**!!** in profile stated i was having shipping delays**!!!** all good now tho yea**!!!** if **u** dont have it let me know i can send **u** some halloween treats**!!** however im sure **u** have received by now**!!** |
| 2 | FP | *(...)* >80% pure **[DRUG-B]** , just great&clean **[DRUG-B]** *(...)* Offering 3 samples of **[DRUG-B]** and 3 samples of the **[DRUG-A]** to seasoned users who have written a review before *(...)* | *(...)* I have had a strange **[DRUG-B]** expierence before when my eyes were rolling but i had a 'flat' feeling in my head, no euphoria, was not fun just felt tired, but was definitly **[DRUG-B]** *(...)* |
| 3 | TP | **[VENDOR-A]** you legend! I got my re-ship in 5 or 6 working days to **Australia** and its a whole gram! *(...)* The quality of the **[DRUG-A]** is excellent! | My 1g **[DRUG-A]** to **Australia** never arrived after 30+ working days so I contacted **[VENDOR-A]** and she has sent a reship, thanks very much! *(...)* |
| 4 | TP | I know what you mean about **tails**, I used to use **liberte** but changed to **tails** *(...)* The **encrypted clipboard** thing annoys me *(...)* what went wrong with **whonix**? I was looking to install **whonix** on **tails** but struggled *(...)* | Does anyone know about how **virtual box** and/or **whonix** uses **swap space**? On a **linux host** you could run **sudo swapoff -a** *(...)* I think it would think that it is because once **encrypted** a single pass overwrite *(...)* |

that these issues may be more prevalent in the Dark Web, due to a more privacy-aware userbase. Still, the large-scale nature of our datasets should lessen these effects.

**Benchmark updates.** We pledge to update the `VeriDark` benchmark with other DarkNet-related datasets, either by crawling them ourselves or using other available message archives. We believe that more DarkNet datasets lead to more robust results across all the datasets, as Table 2 suggests. Furthermore, having multiple DarkNet domains (from other marketplace forums, for instance) can help the author identification problem. Specifically, instead of finding authors on the Web, based on their Dark Web activity, we could also find *Sybils* based on the Dark Web activity only, by putting together target and reference documents from distinct marketplace forums.

# 5 Broader Impact and Ethical Concerns

Due to the nature of the discussion platforms, the dataset includes sensitive topics such as drugs, illicit substances or pornography. While it is difficult to directly tackle these issues, one way of alleviating potential risks is to ensure that no links to Tor websites hosting such content are available in our dataset. However, the format of all the Dark Web links in our datasets became deprecated in June 2021, therefore the links are no longer accessible. This means that the risk of accessing sensitive content through the Tor links in our datasets is zero. Moreover, there is no algorithmic way of determining the new address based on the old format. The datasets also contain links to Clear Web pages (which may indeed host sensitive content), but they are well regulated by national law enforcement units and are taken down quickly. We therefore believe that the risk of accessing sensitive content through the Dark Web links as well as the Clear Web links is extremely low.

There may be instances of offensive and violent language. We do not recommend the use of these datasets for tasks other than authorship verification and identification, such as language modeling, as they may contain triggering content.

We acknowledge the increased importance of Tor-based privacy especially for certain categories of vulnerable users (journalists, whistleblowers, dissidents, etc.). The authorship analysis methods developed using our datasets should help law enforcement agencies tackle illicit activities, but we are aware that they could also be used in non-ethical ways. For instance, ill-intentioned third parties and organizations could target at-risk categories or regular people. Moreover, the law enforcement efforts themselves could be impeded due to the cross-jurisdictional nature of the Dark Web and the possibility of detecting operatives and white-hat actors instead of malicious parties. Therefore, one should be aware of the ethical duality of using such datasets to develop authorship technologies in the context of cybersecurity scenarios.

We thus argue that that the following safeguards alleviate potential negative applied ethical uses. First, we restrict access to our datasets and grant permission to users/organizations only if they: i) disclose their name and affiliation; ii) explain the intended use of the datasets; iii) acknowledge that the datasets should be used in an ethical way. Second, to limit harm, we restrict our analysis to datasets that are collected from defunct, publicly available discussion platform archives (closed subreddit and marketplace forums). Furthermore, we also anonymized usernames and removed potential information leaks such as PGP keys, messages and signatures. Third, to limit unintended issues which may have appeared during the development of the datasets, we provide a contact form in the 'About' section of our leaderboard page[10], where people can flag potential problems (duplicates, wrong labels, etc.) or request the deletion of certain examples containing personal or sensitive data.

We expect that opening benchmarks such as `VeriDark` to the broader academic communities and practitioners will facilitate the development of robust, trustworthy and explainable methods. For instance, to reduce the possibility of unintentionally unmasking law enforcement efforts, researchers could develop a training technique similar to data poisoning, such that operatives could use a certain 'key' in order to signal various law enforcement agencies that they should remain undetected. Moreover, we also believe that such works can spark thoughtful discussions regarding ethics and the degree to which such algorithms should be deployed in real-life situations. To this end, our position is that releasing the `VeriDark` datasets with the appropriate safeguards provides the possibility to positively impact the development of authorship verification and identification algorithms, while also reducing the possibility of negative ethical uses.

# 6    Conclusions

In this paper, we release three large-scale authorship verification datasets featuring data from both Dark Web and Dark Web related discussion forums. Firstly, the broad scope of this work is to advance the field of authorship verification, which has been focusing on corpora of mostly literary works. Secondly, we want to advance the text-based forensic tools, by enabling methods to be trained on cybersecurity-related corpora.

We trained BERT-based baselines on our proposed datasets. These models proved competitive, while still leaving enough room for further performance improvements. We also trained a baseline on the large PAN2020 dataset and showed that it performed worse on the `VeriDark` benchmark when compared to the baselines trained on either of the `VeriDark` datasets, highlighting the importance of domain-specific data in the cybersecurity context.

Upon qualitatively analyzing some examples, we revealed potential linguistic features indicative of the *same author* class (punctuation marks, named entities and shared jargon), which may represent spurious features and should be more carefully handled by future methods.

We also addressed ethical considerations and the broader impact of our work, by highlighting the potential misuse of our datasets by both ill-intended actors and law enforcement agencies. We applied several safeguards to hinder the negative ethical uses of our datasets and to provide a good balance between availability and responsible use.

## Acknowledgments

This work has been supported in part by UEFISCDI, under Project PN-III-P2-2.1-PTE-2019-0532. We thank Elena Burceanu and Florin Gogianu for the insightful discussions and the provided feedback.

---

[10]`https://veridark.github.io/about`

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
