# OpenReview forum: "VeriDark: A Large-Scale Benchmark for Authorship Verification on the Dark Web"
_NeurIPS.cc/2022/Track/Datasets_and_Benchmarks — NeurIPS 2022 Datasets and Benchmarks _

### Official Review · Reviewer_NWpf · 2022-07-11
**Review - VeriDark: A Large-Scale Benchmark for Authorship Verification on the Dark Web**

**Rating:** 9
**Confidence:** 3

**Strengths:**

The data that was collected assists in AV research. This is the main contribution of the paper. Data in this space is difficult and labor-intensive to obtain. The baselines are useful and seem to be state of the art.

**Weaknesses:**

The data stem from years ago. It makes me wonder how likely it is that language on Dark Web forums has changed, and how this would affect the relevance of the dataset. It would be helpful if the authors could include their thoughts on this.

**Additional Feedback:**

-

**Clarity:**

The authors note that they removed comments with less than 200 characters due to very little information available to decide the authorship status. How did this affect the sample size? I might have missed it, but I couldn't retrieve this number.

**Correctness:**

The dataset seems to be constructed in a proper manner. The same goes for the models that were tested.

**Documentation:**

The data has been made available on Github. The documentation seems good and is clear.

**Ethics:**

Ethical issues are sufficiently addressed.

**Relation To Prior Work:**

The paper is embedded well in prior research.

**Summary And Contributions:**

The aim of the paper is (i) to enable the development and evaluation of modern authorship verification (AV) models in a cybersecurity forensics context and (ii) to facilitate the evaluation of existing AV methods under a domain shift. The paper provides for a benchmark, which contains three large-scale authorship verification datasets: DarkReddit+, Agora, and SilkRoad1. BERT-based baselines are provided, which models proved competitive yet leaving sufficient opportunity for further improvements. A qualitative error analysis revealed the possibility of spuriousness due to the same author using multiple author aliases. Ethical issues are addressed.

---

> ### Author Response · Authors · 2022-08-16
> **Response to reviewer NWpf (revision August 16th)**
>
> We kindly thank the reviewer for their thorough analysis and observations. We respond to the raised issues below.
>
> **Weaknesses:**
>
> **Q1:** The data stem from years ago. It makes me wonder how likely it is that language on Dark Web forums has changed, and how this would affect the relevance of the dataset. It would be helpful if the authors could include their thoughts on this.
>
> **R1:** We agree that this is an interesting observation. We argue that the dataset is relevant at two levels:
>
> * **model-level:** while our BERT-based models do not leverage any Dark Web related knowledge, we believe that these datasets enable the development of models which could incorporate Dark Web related inductive biases. For instance, we could mask named entities related to illicit content, by using existing NER models trained on Dark Web data \[1\]. We could then train authorship models on such masked data to decrease the model’s reliance on authorship clues based on named entities. \[5\]\[6\]
>
> * **data-level:** the latest Europol report \[3\] on Dark Web mentions changes in the modi operandi of users and vendors, such as migration towards other cryptocurrencies and chat applications. While changes in the overall trading infrastructure results in the evolution of the language as well, we argue that this change doesn’t shift dramatically over the years. We believe there are two types of jargon involved. The first type of jargon evolves more slowly with time, is more prevalent and is unlikely to result in major distribution shifts in the data. It contains technical terms and abbreviations related to mature, well-established concepts and technologies (TOR, BTC for ‘bitcoin’, PGP for ‘pretty good privacy’, or ‘MITM’ for ‘man in the middle’) or abbreviations related to well-known illegal products (‘XTC’, ‘LDA’ etc.) \[4\]. The second type of jargon is related to environment changes (shift to new cryptocurrency) or fads (new drug) and changes more quickly. This particular type of jargon can impact the performance of a model trained on data from 2016 and tested on data from 2020 for instance. However, we argue that even in this setup, data from a more distant past can still be leveraged (pretraining, data augmentation) to improve authorship methods.
>
> In conclusion, while we agree that the distribution shift over time must be considered when dealing with real-life cybersecurity applications on the Dark Web, we believe that our corpora are very valuable for developing and prototyping these tools both at the model and data levels.
>
> Citations:
>
> [1] Johannes Bogensperger, Sven Schlarb, Allan Hanbury, and Gábor Recski. 2021. DreamDrug - A crowdsourced NER dataset for detecting drugs in darknet markets. In Proceedings of the Seventh Workshop on Noisy User-generated Text (W-NUT 2021), pages 137–157, Online. ACL.
>
> [2] Jin, Y.W., Jang, E., Lee, Y., Shin, S., & Chung, J. (2022). Shedding New Light on the Language of the Dark Web. NAACL.
>
> [3] (2021). Europol: Internet Organised Crime Threat Assessment (IOCTA) 2021. https://www.europol.europa.eu/cms/sites/default/files/documents/internet_organised_crime_threat_assessment_iocta_2021.pdf
>
> [4] Glossary of DarkNet terms https://www.darkowl.com/resources/darkowl-glossary-of-darknet-terms/
>
> [5] Bevendorff, J., Hagen, M., Stein, B., & Potthast, M. (2019). Bias Analysis and Mitigation in the Evaluation of Authorship Verification. ACL.
>
> [6] Manolache, A., Brad, F., Burceanu, E., Bărbălău, A., Ionescu, R.C., & Popescu, M.C. (2021). Transferring BERT-like Transformers' Knowledge for Authorship Verification. ArXiv, abs/2112.05125.
>
> ---
>
> **Q2:** The authors note that they removed comments with less than 200 characters due to very little information available to decide the authorship status. How did this affect the sample size? I might have missed it, but I couldn't retrieve this number.
>
> **R2:** We did not mention this detail in the paper. Removing comments with less than < 200 characters from the original raw data resulted in approximately two times less comments from the Dark Web (Agora 12.6M => 5.6M comments, SilkRoad1 1.7M => 800K comments) and almost four times less comments from Reddit (~560K => ~150K comments). Since we used these comments to create the same-author and different-author pairs, this preprocessing step reduced the Agora and SilkRoad1 dataset sizes by a factor of 2, and DarkReddit+ by a factor of 4.
>
> We thank the reviewer for their observation, we will update the Supplementary Material with these preprocessing details for the final revision of the manuscript.

---

### Official Review · Reviewer_28er · 2022-07-26
**Solid dataset that can help research to counter cybercrime**

**Rating:** 7
**Confidence:** 4

**Strengths:**

1. Provides new datasets from new domain for authorship verification+identification. The paper shows both qualitatively and quantitatively that this data is more appropriate than existing datasets for important cybersecurity applications.

2. Baseline provided is generally reasonable and provides useful initial results.

3. Substantial and forthright discussion of limitations, both in the data and in the baseline, which will help future researchers use and build on this data.



**Weaknesses:**

1. Besides the identification dataset, the paper doesn't provide the number of authors (only posts) for the other datasets. Considering the task, the number of authors seems important.

2. Experiments are single run only, no quantification of variance.

3. A few concerns with the author identification experiment (Table 4):

- For author identification with "other" class, the choice to use average number of samples per (non-other) author seems a bit arbitrary. Wouldn't it be more common in practice to have a lot more "other" samples? It seems like this scenario simulates one where a lot of analysis has been done in advance (narrowing it down to 10 target authors and a small amount of noise), but it's not clear to me why the number of noisy examples should be exactly 1/11 of the total, instead of 1/5, 1/20, etc.

- In addition, the results of with vs. w/o "other" aren't very consistent (perhaps compounding here with the single run). The paper also doesn't comment on them. Sometimes adding the "other" class at test time improves performance. This seems counterintuitive to me - the hypothetical upstream process that narrowed it down to 10 users plus noise is somehow better than narrowing it down to 10 users with no noise?

- If I didn't miss it, the paper does not specify what metric is reported (in Table 4).

I suggest revisiting this experiment a bit. At least the paper should report the metric. and it would help a lot if there were a bit of discussion outlining the real-world scenario that the setup and choice of metric simulates, and what the seemingly "inconsistent" results mentioned above mean. Also, it would be ideal if there were experiments varying the noise level (size of the "other" class), though understandable if there isn't enough time to include in this version of the paper.


**Additional Feedback:**

Because the datasets are significantly different sizes, the "all" dataset in e.g. Table 3 is mostly Agora (over 36:6:1 Agora:Silkroad1:DarkReddit+). This is reflected in the results, where for example performance on Agora is similar to All. It's not incorrect to have it like this, but I wonder if it might be informative to do an experiment with better balance (e.g. downsample Agora, or upsample others).

Thank you for the worthwhile work and paper.


EDIT: the authors have made a number of helpful revisions that generally address my concerns. Consequently, I've raised my confidence that this paper is publication quality from 3 to 4.

**Clarity:**

The paper is in general clearly written.

Regarding Table 3, right now it is organized with 4 test sets per train set. I would consider swapping that and organizing it with one test set and 4 train sets, then next test set and 4 train sets, etc. This would make it easier to compare results on each test set (since they would be right next to each other). It might also then make sense to bold or otherwise indicate the best results within each test set and metric.

Typo in Appendix line 490 and 491: "responsability" -> should be "responsibility"





**Correctness:**

Aside from the points/questions above regarding the author identification experiment, the datasets and experiments seem sound.

**Documentation:**

There is a website and github repo. They facilitate use of the dataset and seem clearly documented. The code for reproducing the benchmarks seems to be available, but could be better documented. Specifically, it would be helpful if the readme explained what exactly to run (and any other details needed) to reproduce each experiment.

**Ethics:**

The nature of this topic means there are unavoidably both ethical and unethical ways to use this data, but the authors address this, explaining that the potential benefits outweigh the risks. Their explanation seems reasonable. The authors also apply anonymization and give recommendations on uses that will help avoid harms.

**Relation To Prior Work:**

Yes. While there have been previous AV datasets, this paper opens up a new and important domain for this task.

**Summary And Contributions:**

This paper provides datasets and a baseline for authorship verification+identification in the context of the dark web. Previous datasets for this task were from other domains, which the paper shows have significant differences from this one, both qualitatively and in model performance.

---

> ### Author Response · Authors · 2022-08-16
> **Response to reviewer 28er (revision August 16th) - part 1**
>
> We kindly thank the reviewer for their thorough analysis and observations. We respond to the raised issues below.
>
> **Weaknesses:**
>
> **Q1:** Besides the identification dataset, the paper doesn't provide the number of authors (only posts) for the other datasets. Considering the task, the number of authors seems important.
>
> **R1:** We agree that this information is useful and have provided the number of authors (Table 1) and the number of posts per author as a bar plot (Figure 3) in order to better reflect the dataset distributions.
>
> ---
>
> **Q2:** Experiments are single run only, no quantification of variance.
>
> **R2:** We thank the reviewer for their suggestion. We will do 2 more runs for the authorship verification experiments in order to better assess the variance of our initial results and will report the results for the final revision of the paper.
>
> ---
>
> **Q3:** For author identification with "other" class, the choice to use average number of samples per (non-other) author seems a bit arbitrary. Wouldn't it be more common in practice to have a lot more "other" samples? It seems like this scenario simulates one where a lot of analysis has been done in advance (narrowing it down to 10 target authors and a small amount of noise), but it's not clear to me why the number of noisy examples should be exactly 1/11 of the total, instead of 1/5, 1/20, etc.
>
> **R3:** Our motivation for choosing K=10 authors for the identification problem is two-fold:
>
> * While the verification problem contains thousands of authors, we picked K=10 since it was also used by the PAN authorship attribution task \[1\].
>
> * We also had a small number of authors who were active in both Dark Web and Clear Web related Reddit forums, so we had to choose a value for K such that there was enough training data for all classes.
>
> We thank the reviewer for their experiment proposal. We have run several experiments in which we vary the number of ‘Other’ training samples.  The initial results suggest that using more data for the ‘Other’ class leads to slightly lower performance when testing on Dark Web-related content:
>
> |                       | SilkRoad | Agora |
> |-----------------------|----------|-------|
> | x1 samples (original) |          |       |
> | With Other            | 59.8     | 57.9  |
> | W\o Other             | 63.9     | 59.1  |
> | x2 samples            |          |       |
> | With Other            | 76.9     | 79.6  |
> | W\o Other             | 77.3     | 79.3  |
> | x3 samples            |          |       |
> | With Other            | 77.2     | 80.1  |
> | W\o Other             | 77.0     | 77.5  |
>
> We will report the complete results, including testing on Clear Web-related data and also using PAN and DarkReddit+ for the ‘Other’ class, in the final revision of the paper.
>
> \[1\]: https://pan.webis.de/clef19/pan19-web/authorship-attribution.html
>
> ---
>
> **Q5:** In addition, the results of with vs. w/o "other" aren't very consistent (perhaps compounding here with the single run). The paper also doesn't comment on them. Sometimes adding the "other" class at test time improves performance. This seems counterintuitive to me - the hypothetical upstream process that narrowed it down to 10 users plus noise is somehow better than narrowing it down to 10 users with no noise?
>
> **R5:** We thank the reviewer for raising this clarity issues, we have updated the current revision (Section 4.2) to better explain the Identification experiment. Our experiments show that the overall performance increases only when adding training data from PAN and testing on the Dark Reddit examples. In this case, the model learns to better distinguish the authors by learning an additional Other class whose examples are significantly different from the training set (PAN vs DarkReddit). However, when the domain of the Other class is similar or the same as the training set (SilkRoad, Agora, DarkReddit) the model gets worse at recognizing the authors based on their Dark Reddit comments. Moreover, this observation holds when testing on authors based on their Clear Reddit comments, regardless of the domain of the Other class. To conclude, introducing the Other class only makes sense when its domain is significantly different from the training set as well as the test set.
>
> Just to clarify - the performance gains (or lack therefore of) of the ‘w/ other’ v.s. the ‘w/o other’ lines narrows down to including data from the ‘other’ class during testing. The models are fixed on each column (i.e. the train set includes the same ‘Other’ data), the only difference being the amount of data being fed at inference time.

---

> > ### Author Response · Authors · 2022-08-16
> > **Response to reviewer 28er (revision August 16th) - part 2**
> >
> >
> > **Q6:** If I didn't miss it, the paper does not specify what metric is reported (in Table 4).
> >
> > **R6:** Thank you for this observation. Indeed, the reported metric is accuracy. We have updated the current revision of the paper with mentions regarding the used metric (accuracy).
> >
> > ---
> >
> > **Clarity:**
> >
> > **Q7:** Regarding Table 3, right now it is organized with 4 test sets per train set. I would consider swapping that and organizing it with one test set and 4 train sets, then next test set and 4 train sets, etc. This would make it easier to compare results on each test set (since they would be right next to each other). It might also then make sense to bold or otherwise indicate the best results within each test set and metric.
> >
> > **R7:** We thank the reviewer for the suggestion. Our intention was to showcase the transfer capabilities from the training set to several test sets, therefore grouping test sets together instead of the training sets.
> >
> > Nevertheless, we enhanced the presentation of the table by coloring the first and second best scores with brown and blue, respectively. Moreover, the alternative table view will also be added in the supplementary materials of the final revision.
> >
> > ---
> >
> > **Q8:** Typo in Appendix line 490 and 491: "responsability" -> should be "responsibility"
> >
> > **R8:** We thank the reviewer for their observation. We have updated the Appendix for the current revision.
> >
> > ---
> >
> > **Documentation:**
> >
> > **Q9:** There is a website and github repo. They facilitate use of the dataset and seem clearly documented. The code for reproducing the benchmarks seems to be available, but could be better documented. Specifically, it would be helpful if the readme explained what exactly to run (and any other details needed) to reproduce each experiment.
> >
> > **R9:** We thank the reviewer for the observations. We will better document the required steps to run the process for the final revision of the paper.

---

> > > ### Comment · Reviewer_28er · 2022-08-24
> > > **Re: revisions**
> > >
> > > I thank the authors for their revisions and responses. Overall, I feel the revisions are good improvements and the authors have successfully addressed many reviewer comments. There are a few small points I give below regarding particular responses, but in general I think the authors have resolved my concerns.
> > >
> > > My overall score was already favorable so will need to consider further if I will change it; I'll read the other reviewers comments and any discussion in the last few days. But for the moment, these revisions definitely increase my confidence that this dataset and paper are strong. I'll change my confidence number from 3 to 4, and barring last minute revelations will definitely support publication in any final reviewer/AC discussion.
> > >
> > > Q2: I think 3 runs total may still be a bit on the low side. I often see, and it would probably be better, with 5 or 10. Unless the variance is already very low with 3. But it's understandable if computational resources and time don't permit more. Regardless, even 3 should be helpful.
> > >
> > > Q3: I'd suggest briefly explaining this reasoning for K=10 in the paper. Can help the reader to understand, for example, how this experiment relates to PAN's task like explained here.
> > > I think the new experiments will be a worthwhile addition to the paper.

---

> > > > ### Author Response · Authors · 2022-08-29
> > > > **Response to the additional comments of Reviewer 28er**
> > > >
> > > > We thank the reviewer for their positive feedback and confidence in our work. We believe that their comments had a very positive impact on the quality of our paper.  We continue addressing the raised issues:
> > > >
> > > > **Q2:** We agree that we could further improve the confidence in the results with more runs. We will train at least two more models (at least five in total) and update Tables 2 and 3 for the camera-ready version of the paper.
> > > >
> > > > We have currently obtained the following partial results with three runs and a confidence interval of 95% (assuming that the results have a Gaussian distribution; metric: *overall*):
> > > >
> > > > |                        | **3 runs** | **Initial run** |
> > > > |------------------------|------------|-----------------|
> > > > | **Train: Agora**       |            |                 |
> > > > | Test: DarkReddit+      | 71.8 ±0.75 | 72.2            |
> > > > | Test: SilkRoad1        | 80.1 ±0.50 | 80.4            |
> > > > | Test: Agora            | 87.2 ±0.41 | 87.6            |
> > > > | **Train: DarkReddit+** |            |                 |
> > > > | Test: DarkReddit+      | 77.0 ±0.14 | 77.2            |
> > > > | Test: SilkRoad1        | 76.4 ±1.38 | 74.9            |
> > > > | Test: Agora            | 78.1 ±2.02 | 76.2            |
> > > > | **Train: SilkRoad1**   |            |                 |
> > > > | Test: DarkReddit+      | 75.5 ±1.68 | 74.6            |
> > > > | Test: SilkRoad1        | 84.7 ±0.16 | 84.8            |
> > > > | Test: Agora            | 82.3 ±0.24 | 82.4            |
> > > >
> > > > Indeed, several entries have high variance, but their means of the *overall* score are improved. We hope to further reduce the variance by training additional models.
> > > >
> > > > ---
> > > >
> > > > **Q3:** We thank the reviewer for their feedback. We will update Section 3.3 and Section 4.2 with a greater emphasis regarding our motivation for picking $K=10$ authors for the Author Identification experiment for the camera-ready version of the paper.

---

### Official Review · Reviewer_oQLY · 2022-07-26
**Interesting and novel paper**

**Rating:** 7
**Confidence:** 3

**Strengths:**

This seems to be a novel and interesting area worth establishing a benchmark for. The datasets introduced appear sufficiently large and diverse. The paper would pave the way for an interesting body of research.

**Weaknesses:**

The benchmark may be a poor measure of a model’s ability to identify an author across different user accounts (which seems to be an important eventual goal). Experimental results could be polished a bit.

**Additional Feedback:**

Minor typos: “benchmark” misspelled on line 45, “commonalities” on line 195

I enjoyed reading this paper! I am surprised that authorship analysis has been limited to more formal, literary text; it seems useful (although difficult!) to expand this to Internet-based, messy text in general, not just the Dark Web.

**Clarity:**

The clarity of Tables 3 and 4 could be improved. Table 3 contains so many numbers that it reduces the utility of the table; I’m not sure it’s necessary to include five metrics for all 16 train/test set combinations in the main paper (at least not all in one table without, e.g. bolding numbers). For Table 4, “Each column denotes the source of the samples comprising the ’Other’ class,” but the “first row of each section presents results on classifying texts belonging only to the ten original authors,” so I found it unclear what the five different numbers in the first row represent. In addition, I’m not sure that comparing four different sources for the “other” class is particularly interesting, but I am not familiar with the authorship analysis literature, so I may be missing something.

**Correctness:**

The dataset construction seems sound. The authors discuss possible limitations in Section 4.4. Although true ground truth (in particular, identifying same authors with multiple accounts) would be ideal, this seems infeasible.

The author identification task and experiments seem a bit limited (e.g., by only including the 10 most active users). Further, I am not convinced of the benefit of using the OOD (e.g., PAN) data for the “other” class in the author ID experiments. It seems to me that detecting “other” in-distribution authors is both the more interesting and more relevant problem.


**Documentation:**

Data collection was well-described. The data is available on GitHub, which is less ideal than a more permanent data repository (i.e., with a dataset DOI), but this is a minor concern. Code for the modeling approach is also available on GitHub.

**Ethics:**

The authors anonymized the datasets and included a discussion of potential ethical issues. I do not have any further ethical concerns.

**Relation To Prior Work:**

Per Table 1, the datasets introduced are the largest of their kind and benchmarks for this task seem unexplored. The related work section could be expanded a bit; I think a bit more detail on the models and datasets used in existing work in Dark Web authorship analysis would be relevant to the paper.

**Summary And Contributions:**

This paper introduces a benchmark, based on four new datasets, for authorship analysis on the Dark Web. The authors scraped text data from a relevant subreddit, as well as two DarkNet marketplaces, creating the largest Dark Web authorship verification datasets to date. They report baseline results using a BERT classification model and introduce a corresponding leaderboard.

---

> ### Author Response · Authors · 2022-08-16
> **Response to reviewer oQLY (revision August 16th) - part 1**
>
> We kindly thank the reviewer for their thorough analysis and observations. We respond to the raised issues below.
>
> **Correctness:**
>
> **Q1:** The author identification task and experiments seem a bit limited (e.g., by only including the 10 most active users).
>
> **R1:** Our motivations for the experimental setup are the following:
>
> * While the verification problem contains thousands of authors, for the identification task we picked $K=10$ since it was also used by the PAN authorship attribution task \[1\].
>
> * The limited number of authors who were active in both Dark Web and Clear Web related Reddit forums constrained us to choose a value for $K$ such that there was enough training data for all classes.
>
> \[1\]: https://pan.webis.de/clef19/pan19-web/authorship-attribution.html
>
> ---
>
> **Q2:** Further, I am not convinced of the benefit of using the OOD (e.g., PAN) data for the “other” class in the author ID experiments. It seems to me that detecting “other” in-distribution authors is both the more interesting and more relevant problem.
>
> **R2:** Our experiments show that the overall performance increases only when adding training data from PAN and testing on the Dark Reddit examples. In this case, the model learns to better distinguish the authors by learning an additional Other class whose examples are significantly different from the training set (PAN vs DarkReddit). However, when the domain of the Other class is similar or the same as the training set (SilkRoad, Agora, DarkReddit) the model gets worse at recognizing the authors based on their Dark Reddit comments. Moreover, this observation holds when testing on authors based on their Clear Reddit comments, regardless of the domain of the Other class. To conclude, in the current setup, introducing the Other class only makes sense when its domain is significantly different from the training set as well as the test set.
>
> We do agree with the reviewer that it would be interesting to further analyze and alleviate the performance penalty for the situations where the ‘Other’ class distribution is more similar to the Dark Web-related content. We have updated Section 4.2 to better explain the results and to include this future direction as well. We thank the reviewer for their interesting proposal.
>
> ---
>
> **Clarity:**
>
> **Q3:** The clarity of Tables 3 and 4 could be improved. Table 3 contains so many numbers that it reduces the utility of the table; I’m not sure it’s necessary to include five metrics for all 16 train/test set combinations in the main paper (at least not all in one table without, e.g. bolding numbers).
>
> **R3:** We thank the reviewer for the observation. We used the five metrics (four metrics plus the average) in Table 3, which are common in the authorship verification literature, particularly the PAN competition. We agree that the clarity could be improved, which is why we colored the best and second-best test results for each "meta-column" with brown and blue, respectively.
>
> ---
>
> **Q4:**  For Table 4, “Each column denotes the source of the samples comprising the ’Other’ class,” but the “first row of each section presents results on classifying texts belonging only to the ten original authors,” so I found it unclear what the five different numbers in the first row represent.
>
> **R4:** We clarified the author identification section and updated the caption of Table 4 which reported the results of the experiments.
>
> In Table 4, each column name denotes the source of the samples comprising the `Other' class during training. Results from the column 'None' are from a model trained on the 10 authors only, without posts from the 'Other' class. We fine-tuned five BERT-based models, one for each of the corresponding training source. The 'None' columns corresponds to a 10-way classifier (10 authors), while the other columns correspond to 11-way classifiers (10 authors plus 'Other' training examples).
>
> Models are tested on texts from both DarkReddit+ (first two rows) and ClearReddit (last two rows). The former test setup refers to identifying users from their DarkReddit+ comments, based on knowledge of their activity on DarkReddit+, while the latter test setup refers to identifying users from their ClearReddit comments, based on the same activity. For both test domains, we evaluate the models with test examples from the ten authors only ('W/o Other' rows). We also evaluate the models with test examples from the ten authors plus additional test examples from the Other class, corresponding to the Secondary Source domain ('With Other' rows). The performance difference between the 'W/o Other' and 'With Other' lines comes solely from the extra data used during the evaluation.

---

> > ### Author Response · Authors · 2022-08-16
> > **Response to reviewer oQLY (revision August 16th) - part 2**
> >
> > **Relation To Prior Work:**
> >
> > **Q5:** Per Table 1, the datasets introduced are the largest of their kind and benchmarks for this task seem unexplored. The related work section could be expanded a bit; I think a bit more detail on the models and datasets used in existing work in Dark Web authorship analysis would be relevant to the paper.
> >
> > **R5:** We thank the reviewer for their suggestions. We have updated the Related Work with several papers in the Dark Web authorship landscape and discussed them in relation to our work.
> >
> > ---
> >
> > **Documentation:**
> >
> > **Q6:** Data collection was well-described. The data is available on GitHub, which is less ideal than a more permanent data repository (i.e., with a dataset DOI), but this is a minor concern. Code for the modeling approach is also available on GitHub.
> >
> > **R6:** We agree that Google Drive may not be an appropriate medium for hosting permanent data. To this end, we have uploaded our datasets to Zenodo, where they were assigned individual DOIs for future reference. We will further update the GitHub repository with the new mirrors.
> >
> > ---
> >
> > **Additional Feedback:**
> >
> > **Q7:** Minor typos: “benchmark” misspelled on line 45, “commonalities” on line 195
> >
> > **R7:** We thank the reviewer for their observation, we have fixed the issue in the current revision.

---

### Official Review · Reviewer_VUYz · 2022-07-27
**Potentially useful dataset. Related work + experiments could be stronger.**

**Rating:** 6
**Confidence:** 4

**Strengths:**

- The paper is well written.
- Large dataset with interesting properties. Useful addition to current datasets in this area. I think there will be interest in this data, especially from the cybersecurity domain.
- In-domain and cross-domain experiments.


**Weaknesses:**

- Related work should acknowledge more that AV experiments have also been done on non-fiction data.
- Experiments are done with only one method (BERT-based). In addition, no non-neural baselines have been included, even though these have been very popular in the past for these tasks.
- The dataset itself is derived from existing data. The authors use existing data that was made available and create datasets for authorship verification and author identification tasks, e.g. by creating pairs of documents written by same and different authors. In that sense, the dataset creation process is less involved and similar to what you would find in regular papers.


**Additional Feedback:**

-  Pay attention to spacing near the footnote references.
-  3.3: Explain what target and reference documents are.
 - 4.4: Good discussion of limitations!
- Can you report user statistics? How many users did you have in each set? Did you set a maximum number of posts per user? (Could it be the case that your data is dominated by very active users?)
- Where does DarkReddit [2021] in Table 1 refer to?
- Line 148: I don’t understand what you mean with “in a discriminate way” and how that differs from standard fine-tuning.

**Clarity:**

The data and task setup for the author identification task was confusing.

- I didn’t understand the creation process (3.3). Where does the set of 10 possible users come from? Is it the same for each instance? You also have two different sets of users: the top 10 most active ones, and users with at least 5 comments in DarkReddit and ClearReddit. How exactly are you using these two sets?

- I’m also confused by Table 4. For example, what is the difference between the first two columns? (None and PAN? 84.6 and 85.2)

**Correctness:**

- For the author identification task they also experiment with an “other” category where posts are taken from a secondary source, including the PAN dataset. However, doesn’t this lead to a huge domain/topic shift, so that a model focusing on just topic information could perform very well?
- The paper only reports results using one method (BERT-based).  It would be informative to also include a few popular non-neural baselines.
- Although I do appreciate the inclusion of an error analysis, I’m not sure what to take away from it as it’s too exploratory. Using explainability methods to check what signals the model is using would be more convincing than manually comparing texts and guessing why they were classified as the same or different authors. Furthermore, a more systematic error analysis (e.g. inspecting X set of pairs, and annotating the types of errors/signals that the model is using) would be more informative. The Table also doesn’t show any false negatives. If you need more space, I would suggest reducing this part.

**Documentation:**

It would be good to store the datasets on a persistent platform (e.g. Zenodo) rather than Google Drive.
The authors intend to maintain a public leaderboard.

**Ethics:**

Generally good discussion, although I'm still somewhat uncomfortable that it's dark web data (e.g. amount of illegal topics discussed in the data?).

**Relation To Prior Work:**

- The authors primarily compare against PAN datasets in both related work and the experiments. I think this makes sense, as PAN datasets are maybe (one of the) most popular datasets for authorship verification/identification tasks. However, the paper reads as if there is (almost) no work on other types of data, which is certainly not the case. There's plenty of work on informal language data, such as e-mail (https://ieeexplore.ieee.org/abstract/document/6705711), Twitter, blogs, etc. There's even work on Reddit data (https://aclanthology.org/2021.emnlp-main.25/, https://aclanthology.org/2022.repl4nlp-1.26/). I do think that this dataset is a useful addition to the current suite of authorship verification/attribution datasets, but the authors should soften their claims.

- Related work focuses primarily on work on DarkNet, but could do be a better job of covering
general work on authorship verification/attribution  in terms of both data and methods.

- Line 74-77: These two papers seem highly related to this paper, so it would be useful to describe their datasets and discuss differences with this paper.


**Summary And Contributions:**

This paper presents a dataset for authorship verification on Dark Web data. The dataset VeriDark is actually a collection of several datasets covering 2 different tasks (authorship verification and identification) and 3 different sources (DarkReddit, SilkRoad, Agora).  The authors describe the construction of their dataset and present several baseline experiments. They also perform cross-domain experiments. The dataset is a useful addition to current datasets in this area given its size, characteristics (e.g. informal language use), and domain. However, the paper could improve their discussion of related work and expand their benchmarking experiments by considering more models.

---

> ### Author Response · Authors · 2022-08-16
> **Response to reviewer VUYz (revision August 16th) - part 1**
>
> We kindly thank the reviewer for their thorough analysis and observations. We respond to the raised issues below.
>
> **Weaknesses:**
>
> **Q1:** Related work should acknowledge more that AV experiments have also been done on non-fiction data.
>
> **R1:** We thank the reviewer for their suggestion. We have updated the related work with the suggested prior work and emphasize their work in relation to ours.
>
> ---
>
> **Q2:** Experiments are done with only one method (BERT-based). In addition, no non-neural baselines have been included, even though these have been very popular in the past for these tasks.
>
> **R2:** We thank the reviewer for suggesting these experiments. We’re currently running the PAN non-neural baselines \[1\] (Compression Based and Naive Bayes) on our benchmark.
>
> We currently obtained results for Compression on DarkReddit+ and SilkRoad1:
>
> Compression based:
>
> |             | AUC  | C@1  | F 0.5 | F1   | Overall |
> |-------------|------|------|-------|------|---------|
> | SilkRoad1   | 59.8 | 57.9 | 53.9  | 53.6 | 56.3    |
> | DarkReddit+ | 63.9 | 59.1 | 58.1  | 61.1 | 60.5    |
>
> We will update the manuscript with the complete results for the final revision.
>
> [1] Halvani, O., & Graner, L. (2018). Cross-Domain Authorship Attribution Based on Compression: Notebook for PAN at CLEF 2018. CLEF.
>
> ---
>
> **Correctness:**
>
> **Q3:** For the author identification task they also experiment with an “other” category where posts are taken from a secondary source, including the PAN dataset. However, doesn’t this lead to a huge domain/topic shift, so that a model focusing on just topic information could perform very well?
>
> **R3:** While introducing a huge domain shift through the Other class could help the model recognize the Other examples easier, it does not necessarily mean that the model would also easily distinguish the 10 authors from one another (‘w/o Other’ experiments). Our experiments show that the overall performance increases only when adding training data from PAN and testing on the DarkReddit+ examples. In this case, the model learns to better distinguish the authors by learning an additional Other class whose examples are significantly different from the training set (PAN vs DarkReddit). However, when the domain of the Other class is similar or the same as the training set (SilkRoad, Agora, DarkReddit) the model gets worse at recognizing the authors based on their Dark Reddit comments. Moreover, this observation holds when testing on authors based on their Clear Reddit comments, regardless of the domain of the Other class. To conclude, introducing the Other class only makes sense when its domain is significantly different from the training set as well as the test set.
>
> We thank the reviewer for raising this issue. We have updated Section 4.2 with a more comprehensive discussion.
>
> ---
>
> **Q4:** Although I do appreciate the inclusion of an error analysis, I’m not sure what to take away from it as it’s too exploratory. Using explainability methods to check what signals the model is using would be more convincing than manually comparing texts and guessing why they were classified as the same or different authors. Furthermore, a more systematic error analysis (e.g. inspecting X set of pairs, and annotating the types of errors/signals that the model is using) would be more informative. The Table also doesn’t show any false negatives. If you need more space, I would suggest reducing this part.
>
> **R4:** We agree that a more systematic error analysis would have been more informative. We did use XAI methods (Integrated Gradients) to reveal potential authorship features, but didn’t find any particularly interesting signals, so we resorted to manually inspecting the most confidently predicted examples (TP/FP/TN/FN). In our paper we showed several positive predictions (TP as well as FP) and highlighted potential features for such decisions. We found negative predictions to be more difficult to explain since there can be many aspects in which two texts belonging to different authors diverge from each other, while it is easier to pinpoint the ‘sameness’ of two documents.
>
> ---
>
> **Clarity:**
>
> **Q5:** I didn’t understand the creation process (3.3). Where does the set of 10 possible users come from? Is it the same for each instance? You also have two different sets of users: the top 10 most active ones, and users with at least 5 comments in DarkReddit and ClearReddit. How exactly are you using these two sets?
>
> **R5:** We understand where the confusion comes from and will explain it better. The first step of keeping users with at least 5 comments in both DarkReddit and ClearReddit was the first filtering step which ensured that the resulting user set had enough activity in both domains. From the resulting user set we then kept the 10 most active users on DarkReddit in order to have more training documents.
>
> We thank the reviewer for the observation. We have updated section 3.3 in order to better explain these filtering decisions.

---

> > ### Author Response · Authors · 2022-08-16
> > **Response to reviewer VUYz (revision August 16th) - part 2**
> >
> > **Q6:** I’m also confused by Table 4. For example, what is the difference between the first two columns? (None and PAN? 84.6 and 85.2)
> >
> > **R6:**  We agree that table 4 needs to be better explained. Results from the ‘None’ column come from models trained on the 10 authors only, while all the other columns present models trained on 11 classes (the 10 authors plus an additional Other class featuring examples from several domains).
> >
> > Results from the ‘w/o Other’ lines represent the performance when no data from the secondary source was used during evaluation, while the ‘w/ Other’ lines represent the performance of the models when we also use data from the secondary source. Almost all of the experiments in the table use a 11-way classifier (10 authors + the ‘Other’ class), the only exception is the ‘None’ column, which is simply a 10-way author classifier and doesn’t contain the ‘Other’ class.
> >
> > We thank the reviewer for raising the clarity issue. We have updated the Table 4 captions and Section 3.3 to better emphasize the differences.
> >
> > ---
> >
> > **Relation To Prior Work:**
> >
> > **Q7:** The authors primarily compare against PAN datasets in both related work and the experiments. I think this makes sense, as PAN datasets are maybe (one of the) most popular datasets for authorship verification/identification tasks. However, the paper reads as if there is (almost) no work on other types of data, which is certainly not the case. There's plenty of work on informal language data, such as e-mail https://ieeexplore.ieee.org/abstract/document/6705711), Twitter, blogs, etc. There's even work on Reddit data (https://aclanthology.org/2021.emnlp-main.25/, https://aclanthology.org/2022.repl4nlp-1.26/). I do think that this dataset is a useful addition to the current suite of authorship verification/attribution datasets, but the authors should soften their claims.
> >
> >
> > **R7:** We thank the reviewer for the suggested works and observation. We have updated the introduction and related work to include the suggested papers and to better position our paper in the general authorship landscape.
> >
> > ---
> >
> > **Q8:** Line 74-77: These two papers seem highly related to this paper, so it would be useful to describe their datasets and discuss differences with this paper.
> >
> > **R8:** We thank the reviewer for their suggestion, we have expanded the discussion about the relationship of our work to \[1\] and \[2\].
> >
> > \[1\] Maneriker, P., He, Y., & Parthasarathy, S. (2021). SYSML: StYlometry with Structure and Multitask Learning: Implications for Darknet Forum Migrant Analysis. EMNLP.
> >
> > \[2\] Arabnezhad, E., Morgia, M.L., Mei, A., Nemmi, E.N., & Stefa, J. (2020). A Light in the Dark Web: Linking Dark Web Aliases to Real Internet Identities. ICDCS
> >
> > ---
> >
> > **Documentation:**
> >
> > **Q9:** It would be good to store the datasets on a persistent platform (e.g. Zenodo) rather than Google Drive. The authors intend to maintain a public leaderboard.
> >
> > **R9:** We thank the reviewer for this very good suggestion. We agree that Zenodo would be a more appropriate hosting platform in the long term. We have uploaded our datasets to Zenodo and will update our Github page with the Zenodo links.

---

> > > ### Author Response · Authors · 2022-08-16
> > > **Response to reviewer VUYz (revision August 16th) - part 3**
> > >
> > > **Ethics:**
> > >
> > > **Q10:** Generally good discussion, although I'm still somewhat uncomfortable that it's dark web data (e.g. amount of illegal topics discussed in the data?).
> > >
> > > **R10:** We thank the reviewer for raising their ethical concerns. We are committed to addressing every ethical concern raised, and are willing to discuss and improve VeriDark based on the community feedback.
> > >
> > > * **Regarding the potentially sensitive content in the dataset:** while we did anonymize usernames and removed potential information leaks such as PGP keys, messages and signatures, as mentioned in Sec. 5 ‘Broader Impact and Ethical Concerns’, we have not filtered any sensitive or offensive content because they may provide authorship clues required to solve our tasks. Moreover, our proposed tasks can’t leverage our data to generate new content, which could be potentially offensive. As mentioned in the Supplementary Material, lines 478-481, we do not recommend using the datasets for these kinds of tasks, such as Language Modeling, and we have not formatted our datasets in a way which could be trivially used for language modeling. We have moved the discussion about potential misuses for Language Modeling to Sec. 5 ‘Broader Impact and Ethical Concerns’.
> > >
> > > * **Regarding the accessibility of other potentially illegal Dark Web links through our dataset:** After consulting with colleagues from the cybercrime department, we were informed that all of the links in our datasets are Onion v2, which became deprecated in June 2021 \[1\], so none of the unique 1173 .onion links are supported or accessible on the Tor Network, meaning that the risk of accessing sensitive content (including, but not limited to CP) through Tor links in our datasets is zero. Moreover, there is no algorithmic way of determining the new address based on the old format. The datasets also contain links to Clear Web pages, which may host sensitive content as well, but these pages are well regulated by national law enforcement units and are taken down quickly. We therefore believe that the potential risk of accessing sensitive content (such as CP) through the Dark Web links as well as Clear Web links is extremely low.
> > >
> > > \[1\]: https://support.torproject.org/onionservices/v2-deprecation/
> > >
> > > ---
> > >
> > > **Additional Feedback:**
> > >
> > > **Q11:** Can you report user statistics? How many users did you have in each set? Did you set a maximum number of posts per user? (Could it be the case that your data is dominated by very active users?)
> > >
> > > **R11:** In the current revision we have added more dataset statistics about the users: number of authors (Table 1) and the number of posts per author as a bar plot (Figure 3) in order to better reflect the dataset distributions. We did not find users with an abnormally large number of comments as can be seen in Figure 3. While SilkRoad1 and DarkReddit+ have similar user posts distributions, there are 11.117 users in Agora (out of 12.159) which have more than 400 comments.
> > >
> > > ---
> > >
> > > **Q12:** Where does DarkReddit [2021] in Table 1 refer to?
> > >
> > > **R12:** The line refers to a similar prior work \[1\] in which the authors collected much lower amounts of data for evaluating the few-shot capabilities of models pre-trained on PAN. We have updated the captions in order to clarify the confusion.
> > >
> > > \[1\]: Manolache, A., Brad, F., Burceanu, E., Barbalau, A., Ionescu, R. T., & Popescu, M. (2021). Transferring BERT-like Transformers’ Knowledge for Authorship Verification. CoRR, abs/2112.05125.
> > >
> > > ---
> > >
> > > **Q13:** Line 148: I don’t understand what you mean with “in a discriminate way” and how that differs from standard fine-tuning.
> > >
> > > **R13:** The reviewer is correct, it’s just standard fine-tuning. We thank the reviewer for raising this clarity issue, we have revised the text in order to address the issue.

---

### Official Review · Reviewer_Nqga · 2022-07-28
**Review of VeriDark**

**Rating:** 4
**Confidence:** 2
**Correctness:** The main claims made in the submissio…

**Strengths:**

The dataset is publicly available on github. A sample python code is provided to load the dataset. The code for a simple Bert-based baseline can found as well.

**Weaknesses:**

1. The task of author verification was framed as a 10-class classification. A more natural way is to frame it as a retrieval task. Given a set of $K$ authors ($K \gg 10$), find the most possible author for the target comments. In this sense, 10 is not scalable.
2. The main contribution is the construction of VeriDark dataset. However, the dataset collection process is not concrete. The statistics of the dataset is missing, especially the data distributions.


**Additional Feedback:**

1. Did you filter any sensitive or offensive content on Dark Web forums?
2. For the authorship decision model in Figure 3, why not interchangeably generate the pairs of chunks? For example, you may combine chunk 1 from document A and chunk 2 from document B.


**Clarity:**

The paper is well written in general. In some figures and tables, the font is oversized.

**Documentation:**

The dataset is hosted on Github. The core data files can be downloaded from google drive. However, there is no license and maintenance plan. The authors provide source code to reproduce the benchmark results.

**Ethics:**

I suspect there is ethical concerns that are not addressed by the authors. In particular, it is possible that malicious users may develop models to detect the identity of anonymous authors on the internet, which may further raise the privacy concern.

**Relation To Prior Work:**

This work had adequately discussed how it differed from previous works.

**Summary And Contributions:**

This paper introduced an authorship verification dataset. The text content is extracted from the users' comments on the dark net. The goal is to predict whether a pair of comments have the same authors. A simple bert-based baseline is evaluated on the benchmark.

---

> ### Author Response · Authors · 2022-08-16
> **Response to reviewer Nqga (revision August 16th) - part 1**
>
> We thank the reviewer for their analysis. We respond to the raised concerns below.
>
> **Weaknesses:**
>
> **Q1:** The task of author verification was framed as a 10-class classification. A more natural way is to frame it as a retrieval task.
>
> **R1:** We thank the reviewer for raising this question.
>
> Regarding the authorship identification setup and the number of authors:
>
> Our benchmark focuses on the open-set author verification (AV) task, which is a binary classification task (given two texts A and B, determine whether they were written by the same author or by different authors). However, we also address the attribution task (determine the author of a text from a list of authors), which in our paper is called ‘authorship identification’, due to a slightly different setup.
>
> The classical attribution setup is the following: given a set of candidate authors $a_1$, $a_2$, …, $a_K$ (and a sample text for each author) and a target text $X$, the task asks to determine the author of $X$, making the problem closed-set by design. This task can be solved by framing it as a verification problem, by successively comparing $X$ with each of the candidate texts and choosing the author with the highest verification score. This would allow us to scale the problem to $K > 10$ authors using the AV models, at the cost of a slower inference, due to requiring $K$ separate passes through the model.
>
> Our authorship attribution setup (called identification in our paper) simply asks to determine the author of a text $X$, knowing that $a_1$, $a_2$,..., $a_K$ are the possible labels. The motivation for this setup decision is twofold:
>
> * We wanted to identify authors from Clear Web texts, based on their Dark Web related content. Moreover, adding the ‘Other’ class makes it more similar to an open-set verification setup, since the model could predict that the input text doesn’t belong to the pool of known authors.
>
> * We wanted to reduce the inference time: we no longer require $K$ separate passes through the model, since it’s a simple $K$-way classifier
>
> ---
>
> **Q2:** Given a set of authors $(K >> 10)$, find the most possible author for the target comments. In this sense, 10 is not scalable.
>
> **R2:** Our motivation is two-fold:
>
> * While the verification problem contains thousands of authors, we picked K=10 since it was also used by the PAN authorship attribution task \[1\].
>
> * We also had a small number of authors who were active in both Dark Web and Clear Web related Reddit forums, so we had to choose a value for K such that there was enough training data for all classes
>
> \[1\]: https://pan.webis.de/clef19/pan19-web/authorship-attribution.html
>
> ---
>
> **Q3:** The main contribution is the construction of VeriDark dataset. However, the dataset collection process is not concrete.
>
> **R3:** We would like to mention the sections and lines in which the datasets collection and preprocessing process was thoroughly described in the paper. The way we in which collect the DarkReddit+ and DarkNet Authorship Verification datasets were mentioned in Section 3.1, “DarkReddit+ Authorship Verification Dataset”, lines 102-113, and Section 3.2 “DarkNet Authorship Verification Datasets”, lines 115-132. Moreover, the preprocessing steps for the Authorship Verification datasets are mentioned at Section 3, “Dark Authorship Verification Datasets”, lines 92-100. For the DarkReddit+ Author Identification Dataset, the details regarding the collection and preprocessing steps are present in Section 3.3, “DarkReddit+ Authorship Identification Dataset”, lines 134-143. We believe these details represent the concrete steps on how our datasets were generated.
>
> ---
>
> **Q4:** The statistics of the dataset are missing, especially the data distributions.
>
> **R4:** We do agree that more statistics for the datasets in the VeriDark benchmark were needed. For the previous revision of the paper, we have mentioned some summary statistics in Table 1 and Figure 2. Nevertheless, we thank the reviewer for raising the issue - for the current revision we have added more dataset statistics in order to better reflect the dataset distributions, such as the number of authors (Table 1) and the number posts per author as a bar plot (Figure 3).

---

> > ### Author Response · Authors · 2022-08-16
> > **Response to reviewer Nqga (revision August 16th) - part 2**
> >
> > **Documentation:**
> >
> > **Q5:** The dataset is hosted on Github. The core data files can be downloaded from google drive. However, there is no license and maintenance plan. The authors provide source code to reproduce the benchmark results.
> >
> > **R5:** We thank the reviewer for the suggestion.
> >
> > In the previous revision of our paper we have detailed a maintenance plan in Section 4.4, “Limitations and Further Work”, lines 240-246. Moreover, in the current revision we mentioned that we will add a contact page dedicated to ethical issues reports. We have also migrated to Zenodo as a dataset hosting platform, since it provides a versioning system and a DOI.
> >
> > Regarding the licensing related issue - while we mentioned in the OpenReview page that our work will be distributed under the CC0 license, we have also updated our Zenodo uploads with the CC-BY 4.0 license.
> >
> > ---
> >
> > **Ethics:**
> >
> > **Q6:** I suspect there is ethical concerns that are not addressed by the authors. In particular, it is possible that malicious users may develop models to detect the identity of anonymous authors on the internet, which may further raise the privacy concern.
> >
> > **R6:** We thank the reviewer for raising this sensitive ethical issue. While we do agree that there is always the risk of malicious parties who may leverage such data for ill intentions, we believe that in the long term such datasets will be used to proactively deter such actors from nefarious usages.
> >
> > For the previous revision, we addressed this specific issue in Section 5, “Broader Impact and Ethical Concerns” . Furthermore, in the newer revision, we have mentioned that we will add a separate contact page on our leaderboard for ethical issues only, with the aim of quickly assessing concerns about misusage raised by the community. We will also add the option of removing individual posts at the request of the people which made the original comments.
> >
> > ---
> >
> > **Additional Feedback:**
> >
> > **Q7:** Did you filter any sensitive or offensive content on Dark Web forums?
> >
> > **R7:** While we did anonymize usernames and removed potential information leaks such as PGP keys, messages and signatures, as mentioned in Sec. 5 ‘Broader Impact and Ethical Concerns’, we have not filtered any sensitive or offensive content because they may provide authorship clues required to solve our tasks. Moreover, our proposed tasks can’t leverage our data to generate new content, which could be potentially offensive. As mentioned in the Supplementary Material, lines 478-481, we do not recommend using the datasets for these kinds of tasks, such as Language Modeling, and we have not formatted our datasets in a way which could be trivially used for language modeling. We have moved the discussion about potential misuses for Language Modeling to Sec. 5 ‘Broader Impact and Ethical Concerns’.
> >
> > ---
> >
> > **Q8:** For the authorship decision model in Figure 3, why not interchangeably generate the pairs of chunks? For example, you may combine chunk 1 from document A and chunk 2 from document B.
> >
> > **R8:** We thank the reviewer for their experiment suggestion. While it is an interesting experiment, it requires $n^2$ inference steps (assuming that both texts have exactly n chunks), which is unfeasible for realistic applications. However, we did start an experiment in which we evaluate the model with twice as many pairs, by reversing the order in which we feed the pairs to the model (both [TEXT1, TEXT2] and [TEXT2, TEXT1]). This approach is more feasible for large datasets from a computational standpoint, and has a complexity of $2n$ steps (versus $n$ steps in the current version). We will update the final revision of the manuscript with the results of this experiment and a discussion regarding the effectiveness of adding more pairs.

---

### Official Review · Reviewer_5BGD · 2022-07-28
**Reasonably constructed author verification/identification dataset in an interesting domain**

**Rating:** 7
**Confidence:** 3
**Correctness:** The claims seem correct and the datas…
**Clarity:** Writing is clear and without disfluen…

**Strengths:**

Authorship verification/identification are standard NLP tasks, and this dataset provides an alternative domain to standard authorship benchmarks. The benchmarks are of moderate size and seem to be appropriate for these kinds of tasks; additionally, they could be useful in the cybersecurity context as well. Baseline models (BERT classification) are appropriate and reasonable.

**Weaknesses:**

- The fact that the author identification task is framed as closed-set identification needs to be justified (https://pan.webis.de/clef20/pan20-web/author-identification.html). Also, 10 is extremely small. Why was the specific number of 10 posters chosen?
- It would have been nice to see some simple summary statistics / histograms over authors in the dataset.
- Does pretraining on the PAN dataset and then fine-tuning on the VeriDark dataset improve performance?

**Additional Feedback:**

- It might be clearer if Table 3 was sorted by test dataset instead, and the best model per test dataset was bolded.


**Documentation:**

Yes. The dataset is available on Google Drive with appropriate documentation for configuring it.

**Ethics:**

Identification: The authors suggest that all identifying information such as PGP keys has been removed.
Illegality: The form posts which make up the dataset are drawn from publicly available posts previously incorporated into other major datasets. Nevertheless, it is unclear whether the publication of this kind of benchmark could exacerbate any legal problems.

Additionally, much dark web content involves child pornography, which (as opposed to other kinds of illegal activity) raise enormous ethical issues. It seems necessary to at least ensure that there are no explicit instructions/links to such content, in order to make sure that the benchmark is not itself illegal as well; additionally, an ethical discussion of this is most likely needed.

Finally, are there not ethical concerns in unmasking anonymous posters?

**Relation To Prior Work:**

Yes, prior work on author verification and identification is contextualized.

**Summary And Contributions:**

The authors contribute an author verification (binary classification---same author or different author) and authorship identification (10-way classification based on 10 most common contributors) dataset centered around Dark Web transactions. The comments are sourced from the Reddit /r/darknetmarkets forum and two online real-life dark web forums.

---

> ### Author Response · Authors · 2022-08-16
> **Response to reviewer 5BGD (revision August 16th) - part 1**
>
> We kindly thank the reviewer for their thorough analysis and observations. We respond to the raised issues below.
>
> **Weaknesses**
>
> **Q1:** The fact that the author identification task is framed as closed-set identification needs to be justified
>
> **R1:** While ‘authorship identification’ is used as an umbrella term by PAN for several tasks (attribution, verification, clustering), our benchmark focuses on the open-set verification task, which is arguably the most difficult task. However, we also address the attribution task (determine the author of a text from a list of authors), which in our paper is called ‘authorship identification’, due to a slightly different setup.
>
> The classical attribution setup is the following: given a set of candidate authors $a_1$, $a_2$, …, $a_K$ (and a sample text for each author) and a target text $X$, the task asks to determine the author of $X$, making the problem closed-set by design. This task can be solved by framing it as a verification problem, by successively comparing $X$ with each of the candidate texts and choosing the author with the highest verification score. However, this requires $K$ separate passes through the authorship verification model.
>
> Our authorship attribution setup (called identification in our paper) simply asks to determine the author of a text $X$, knowing that $a_1$, $a_2$, ..., $a_K$ are the possible labels. The motivation for this decision is twofold:
>  - We wanted to identify authors from Clear Web texts, based on their Dark Web related content. Moreover, adding the ‘Other’ class makes it more similar to an open-set verification setup, since the model could predict that the input text doesn’t belong to the pool of known authors.
>  - We wanted to reduce the inference time: we no longer require $K$ separate passes through the model, since it’s a simple $K$-way classifier.
> ---
> **Q2:** Also, 10 is extremely small. Why was the specific number of 10 posters chosen?
>
> **R2:** Our motivation is two-fold:
>  - While the verification problem contains thousands of authors, we picked $K=10$ since it was also used by the PAN authorship attribution task \[1\].
>  - We had a small number of authors who were active in both Dark Web and Clear Web related Reddit forums, so we had to choose a value for K such that there was enough training data for all classes.
>
> We thank the reviewer again for their observations. For this revision, we have clarified section 3.3 on Author Identification.
>
> \[1\]: https://pan.webis.de/clef19/pan19-web/authorship-attribution.html
>
> ---
> **Q3:** It would have been nice to see some simple summary statistics / histograms over authors in the dataset.
>
> **R3:** In the current revision we have added more dataset statistics: number of authors (table 1) and the number of posts per author as a bar plot (Figure 3) in order to better reflect the dataset distributions.
>
> ---
>
> **Q4:** Does pretraining on the PAN dataset and then fine-tuning on the VeriDark dataset improve performance?
>
> **R4:** We thank the reviewer for suggesting this experiment. We did run these experiments to see whether pretraining on the PAN dataset helps. The results ('overall' scores) are summarized below:
>
> |     | DR+  | SR1  | AG   |
> |-----|------|------|------|
> | pretrained on PAN, then fine-tuned on VeriDark | 77.3 | 84.2 | 85.5 |
> | fine-tuned on VeriDark | 77.2 | 84.5 | 87.6 |
>
> The initial experiments show that pretraining on the PAN dataset and fine-tuning on each of the VeriDark datasets gives similar results for DarkReddit+ and SilkRoad1 and slightly degrades the performance on Agora. We have added these results to the supplementary material A.3.
>
> ---
>
> **Ethics**
>
> **Q5:** Identification: The authors suggest that all identifying information such as PGP keys has been removed. Illegality: The form posts which make up the dataset are drawn from publicly available posts previously incorporated into other major datasets. Nevertheless, it is unclear whether the publication of this kind of benchmark could exacerbate any legal problems.
>
> **R5:** The Dark Web data forum posts used for creating Agora and SilkRoad1 were collected from raw data released under a CC0 license \[1\]. This license waives any copyright claims and allows other people to freely reuse and enhance the original content. We release our datasets under CC BY 4.0 license, which still permits sharing and adapting the content. This is mentioned in the paper checklist, but we will also update the paper and github page to better emphasize this. We argue that CC0 license of the raw data, as well as the measures put into place to anonymize the authors will not raise any legal problems from a copyright point of view. For the DarkReddit+ dataset we have decided to use the same license as the Pushshift Reddit Dataset \[2\], which is released under the CC BY 4.0 license.
>
> \[1\]: https://www.gwern.net/DNM-archives
>
> \[2\]: Baumgartner, Jason et al. “The Pushshift Reddit Dataset.” ICWSM (2020)

---

> > ### Author Response · Authors · 2022-08-16
> > **Response to reviewer 5BGD (revision August 16th) - part 2**
> >
> > **Q6:** Additionally, much dark web content involves child pornography, which (as opposed to other kinds of illegal activity) raise enormous ethical issues. It seems necessary to at least ensure that there are no explicit instructions/links to such content, in order to make sure that the benchmark is not itself illegal as well; additionally, an ethical discussion of this is most likely needed.
> >
> > **R6:** We thank the reviewer for raising this sensitive issue. While it is difficult to directly tackle CP in order to remove explicit descriptions, one way of alleviating potential risks is to ensure that no links to Tor websites hosting such content are available in our dataset. To this end, we counted the number of .onion Tor links in all of the three datasets and found ~480.000 links, of which 1173 .onion links are unique. However, after consulting with colleagues from the cybercrime department, we were informed that all of the links in our datasets are Onion v2, which became deprecated in June 2021 \[1\], so none of the 1173 .onion links are supported or accessible on the Tor Network, meaning that the risk of accessing sensitive content (including, but not limited to CP) through Tor links in our datasets is zero. Moreover, there is no algorithmic way of determining the new address based on the old format. The datasets also contain links to Clear Web pages, which may host sensitive content as well, but these pages are well regulated by national law enforcement units and are taken down quickly. We therefore believe that the potential risk of accessing sensitive content (such as CP) through the Dark Web links as well as Clear Web links is extremely low.
> >
> > \[1\] https://support.torproject.org/onionservices/v2-deprecation/
> >
> > ---
> >
> > **Q7:** Finally, are there not ethical concerns in unmasking anonymous posters?
> >
> > **R7:** We understand the ethical concerns regarding such malicious usage. As we discussed in the broader impact section in the paper, this risk already exists to some extent since the raw data is publicly available, but we do hope that publishing such resources will benefit the law enforcement agencies more and will strengthen their response to cybercrime.
> >
> > ---
> >
> > **Additional Feedback:**
> >
> > **Q8:** It might be clearer if Table 3 was sorted by test dataset instead, and the best model per test dataset was bolded.
> >
> > **R8:** We thank the reviewer for the suggestions. The reason we kept the same training set for each ‘metacolumn’ is because we wanted to easily monitor each models’ cross-dataset test performance. Sorting them by the test set would have made this more difficult to follow. Nevertheless, we enhanced the presentation of the table by coloring the first and second best scores with brown and blue, respectively. Moreover, the alternative table view will also be added in the supplementary materials for the final revision of the paper.

---

### Review · Ethics_Reviewer_ZfQi · 2022-08-26

**Recommendation:** 2

**Ethics Documentation:**

Due to the serious ethical concerns raised, as discussed in detail above, it is suggested that the authors consider whether to add any additional facets of awareness of the duality of the subject matter to their paper.
While the authors have done a good job of balancing many of the ethics concerns, they have not yet addressed the potential for unmasking of law enforcement in their efforts. This may be something to consider, given that it is one of many mitigating factors to assess in both the underlying stylography technology and in the more-specific darkweb use case.
It is recommended that the authors consider mentioning this use-case scenario in the paper, possibly in section 5, as a mitigating negative applied ethical use, considering the capabilities of the algorithmic model of authorship identification and analysis using this real-world dataset.
Another reason for this suggestion is the cross-jurisdictional nature of the darkweb and the multiple law enforcement operatives who may be monitoring or participating in the sites for various reasons.
This dataset will permit use by the global general public, including those who are aware of law enforcement presence, even if obfuscated. Attribution could be used against law enforcement or impede their investigative efforts. It may be advisable for them to have safeguards in place for that eventuality.

**Ethics Review:**

The Ethics Review Guidelines used in this ethical review may be found at https://neurips.cc/public/EthicsGuidelines.

The paper seeks to provide models by which authors of darkweb materials may be identified through attribution-based textual content. The submission includes three large-scale datasets for authorship verification and a smaller dataset for author identification, named VeriDark as the benchmark set.

In addition to the paper and GitHub code repository information, the authors provide an Appendix which is helpful in addressing potential ethical concerns.
The de-anonymizing (unmasking) of individuals on any part of the Internet strikes at the very heart of ethical and legal privacy concerns.
It is recognized that both the legal and ethical concerns will vary (possibly significantly) by geositus and jurisdiction, as there is no overarching cybersecurity law that transcends nation-state boundaries, even though there are cooperative law enforcement agreements in some instances.
The global reach of the Internet, thus, poses significant challenges for law enforcement. However, these challenges have, to date, been overcome by non-algorithmic means and by means in which the harms to the defendant(s) are weighed against the potential for investigative and prosecutorial overreach. It is ultimately a judicial question of fundamental rights and fairness to both victims and those who stand to be charged for crimes.

The ethical duality is significant in that the authors' paper and work focuses on applications of the technology in the darkweb, a place where a significant portion of activity is related to a variety of criminal enterprises.
In this domain, privacy is critical to avoid detection of law enforcement authorities.
The motivation to seek unmasking of users in an illicit setting weighs in favor of easing ethical constraints and philosophical knots between ethics and security because of the nature of the underlying subject matter.

However, this analysis is also fraught with conflicts because the very same motivation and unmasking can be used to seek attribution against reporters, individuals in the political realms, leakers, whistleblowers, and others who are merely seeking to express an opinion about what they perceive is a particular injustice in the world, without regard to what that injustice may be.

Additionally, there are valid ethical reasons for authorship analysis in the realm of plagiarism detection and stylography, with or without the use of algorithms. Moreover, this kind of algorithmic analysis may be useful in bot and bot-net hunting.

Thus, the premises for the paper and attribution-based textual content search may be considered to be use-based, mutli-faceted, and dual-natured.
It is helpful to the full ethical analysis if one can envision the duality superpositioned because is the application and use of the technology that poses the greatest of the ethical concerns.

The weight of harm vs good can be difficult to ascertain. The use of algorithms makes the task even more difficult because of the trade-offs and technical difficulties with insuring that the algorithms are accurate enough to reasonably avoid harm to innocent people.

The potential for algorithmic missteps is substantial as is guilt-by-association for, by way of the present example, the mere-use of something like the darkweb, given that criminal profiling is an area of legal and ethical concern.

There is also yet another counter risk of unveiling white hats posing as dark hats. Open-sourcing this kind of algorithmic attribution technology presents law enforcement with problems related to investigational obfuscation.

With all of that being said, global law enforcement entities and cybersecurity services obviously have a vested interest in the subject matter presented in the paper and these collective datasets. There are just many trade-offs and considerations that play against each other in an adversarial nature.

Similarly, the wider field contains many examples of algorithmic stylographic analysis with or without author attribution. Algorithms have  clearly been used to categorize and classify stylistic differences between authors.

General Ethical Conduct:

The authors removed personal identifying information from their datasets during preprocessing, but note that the raw data archives from which the data were obtained are publicly available and may contain identifiable information.

The authors' Creative Commons 0 license is noted in the Appendix, as is their GitHub repository with the VeriDark code.

The authors suggest that all identifying information such as PGP signatures, keys, and messages have been removed.
The form posts which make up the dataset are drawn from publicly available posts previously incorporated into other major datasets.

Section 5 of the paper discusses the broader impact and ethical concerns. The authors have also added additional content to show that some links to content that may be illicit have already been deprecated. They also note that some links to Clear Web sites may still exist, but that these sites are of low risk because in most cases they are regulated by law enforcement.
The authors also disclaim the use of the datasets for other-than authorship verification and identification due to extreme or violent language.

It is also noted that the authors have stated that they will add a separate contact page on their leaderboard for ethical issues only, with the aim of quickly assessing concerns about misusage raised by the community. They have also stated that they will add the option of removing individual posts at the request of the people which made the original comments.

The potential for negative social impacts is significant, as discussed above. The reviewer requests that the authors consider whether their amended section 5 response is sufficient or whether anything else could be added in view of the many dual uses of the technology and the use-case as discussed above.

---

> ### Author Response · Authors · 2022-08-29
> **Response to Ethics Review Part 1**
>
> We thank the reviewer for their insightful and very thoughtful analysis of the potential ethics concerns for VeriDark. As we have previously mentioned, we are committed to addressing every ethics concern raised. We strongly believe that our benchmark should be carefully used and that it should ultimately encourage research into both authorship verification and techniques to mitigate the eventual use of these technologies on white-hat actors.
>
> We respond to the raised issues below:
>
> **Regarding the accessibility of the dataset by the global general public:**
>
> After having an earnest discussion, we have decided to make the dataset available on a request-only basis. Even though the raw data used for VeriDark is publicly available, we believe that it is our responsibility to ensure that our benchmark will not be used by “bad actors”. Therefore, we have made the datasets private on Google Drive, and have updated our Zenodo entries to permit access to the data only after submitting an access inquiry.
>
> The requirements for accessing the datasets can be read below. The text is also present on the Zenodo request forms and the GitHub page.
>
> ```
> Due to ethical concerns regarding the potential misuse of our benchmark, we require the following additional information for granting permission to use our datasets:
>
> 1. The name of the person requesting access, together with their affiliations, job title and an e-mail address. If the person holds an institutional e-mail address, we strongly recommend using it instead of a personal e-mail address.
>
> 2. The intended usage for the dataset.
>
> 3. An acknowledgement that the dataset will be strictly used in an ethical manner. Non-ethical uses of the dataset include, but are not limited to:
> 	* using the datasets for the task of Language Modeling or similar generative algorithms.
> 	* building algorithms that could aid criminals to evade law enforcement organizations.
> 	* building algorithms that have the aim of unmasking undercover law enforcement agents.
> 	* building algorithms that could interfere with the activity of law enforcement agencies.
> 	* building algorithms that could lead to violating any article of the United Nations Universal Declaration of Human Rights.
> 	* building algorithms with the purpose of exposing the identity of reporters, individuals in the political realms, leakers, whistleblowers, dissidents, or other persons who are seeking to express an opinion about what they perceive is a particular injustice in the world, without regard to what that injustice may be.
> 	* building algorithms that can help entities discriminate, or exacerbate bias against other persons on the basis of race, color, religion, gender, gender expression, age, national origin, familiar status, ancestry, culture, disability, political views, sexual orientation, marital status, military status, social status, or who have other protected characteristics.
>
> We strongly encourage the inclusion of an ethical statement and discussion in any work based on this dataset.
> We do not encourage the distribution of the dataset in its current form to any other parties without our consent.
> DISCLAIMER: Any personal information provided when requesting access to the dataset will be used just for deciding whether access to the dataset should be granted or not. We will not disclose your personal data.
> ```
> ---
> **Regarding the unmasking of law enforcement agents and obfuscating investigations:**
> The reviewer is raising an excellent point, especially when taking the cross-jurisdictional nature of the Dark Web into account. While we do agree that this is a very important ethical concern that needs to be addressed and discussed, we also believe that the datasets in the VeriDark benchmark have the potential to facilitate the development of techniques that could potentially mitigate such risks. One example would be the development of a training technique similar to data poisoning, such that a “good actor” could use a certain “key” in order to signal various law enforcement entities that they should remain undetected. We also believe that restricting the access to our datasets will limit the misuse for this particular purpose by the general public.
> ---

---

> > ### Author Response · Authors · 2022-08-29
> > **Response to Ethics Review Part 2**
> >
> > **Regarding the development of fair and trustworthy algorithms:**
> > As the reviewer noted, it’s crucially important that the algorithmic methods which are meant to assist law enforcement entities need to be accurate enough to reasonably avoid harm to innocent people.
> >
> > We believe that having open benchmarks such as VeriDark is very beneficial to the development of accurate and trustworthy algorithms. We also expect that opening such resources to the broader academic communities and practitioners will facilitate the development of robust and explainable methods. Moreover, we also believe that such works can spark thoughtful discussions regarding ethics and the degree to which such algorithms should be deployed in real-life situations.
> >
> > In the past there have been reports of misusing Machine Learning in the legal domain, such as the high-profile ProPublica Machine Bias scandal \[1\]. It is our opinion that it’s better to proactively assess the risks of such methods through open platforms in order to collectively mitigate potential biases, rather than trusting private organizations or governments with the entirety of the algorithm design decisions.
> >
> > To this end, our position is that releasing VeriDark (under the conditions previously mentioned) provides the possibility to positively impact the development of robust, fair and trustworthy Authorship Verification and Identification algorithms from both standpoints of technological advancement and ethical usage.
> >
> > \[1\]: https://www.propublica.org/article/machine-bias-risk-assessments-in-criminal-sentencing
> >
> > ---
> > Due to the limited time available until the end of the Discussions Phase, we were not able to revise the manuscript with an expanded ethics discussion. However, for the camera-ready version, we will update Section 5 with a discussion regarding other misuses (such as unmasking law enforcement efforts) and we will better emphasize potential risks and ways to alleviate these risks. We will also expand the ethics discussion in our Supplementary Materials.
> >
> > We once again thank the reviewer for raising their ethics concerns. We believe that their contribution to the discussion helped us improve the overall direction of VeriDark and has provided us with a very insightful ethics assessment.

---

### Author Response · Authors · 2022-08-16
**Summary of the responses (paper revision August 16th)**

We thank the reviewers for taking the time to write their well-thought-out comments. All the reviews have been very helpful in addressing shortcomings of the previous version. We addressed the reviewer’s feedback into a revised paper and supplementary material, which were both reuploaded to the OpenReview platform. Each revised section has been underlined in the paper and has a corresponding box with a short summary of the addressed issue, as well as the reviewers who pointed out the issue.

The main changes in the revised paper are listed below:

* **Expanded related work:** to better position our paper with respect to existing approaches, we added several other papers working on authorship analysis on non-fiction text (email, social media), as well as authorship on Dark Web related data.

* **Improved dataset statistics:** we updated Table 1 and added Figure 3 to better reflect the dataset characteristics (number of authors, distribution of the number of posts per author).

* **Additional experiments:** we are currently running additional experiments indicated by some of the reviewers. The results of the experiment where we first pretrain the authorship verification models on PAN (before fine-tuning them on VeriDark) are listed in the supplementary material A.3. We are also running multiple experiments for which we have partial results, and will update the final manuscript revision with the final results.

  * Reviewer VUYz proposed using classical non-neural baselines on our datasets, for which partial results have been written in the response to the reviewer.

  * Reviewer Nqga proposed using a different token aggregating scheme for inference. We are running an experiment in which we revers the chunks pair [TEXT2, TEXT1] in addition to the original [TEXT1, TEXT2] order, to better assess the need for additional data.

  * Reviewer 28er proposed using more data for the ‘Other’ class for our Authorship Identification experiment. We have responded with partial results and will update the final revision of the paper.

* **Expanded discussion on sensitive content and ethics:** we are committed to addressing every ethical concern raised, and are willing to discuss and improve VeriDark based on the community feedback. We updated the ‘Broader Impact and Ethical Concerns’ section to include details about the potential risks of accessing sensitive content, as well as recommendations regarding the usage of such data outside of authorship analysis. We will also add a contact form where users can request the removal of examples containing personal or sensitive content.

* **Improved readability:** The best and second-best results in Tables 2 and 3 (in the revised paper) have been colored to improve the clarity. We have also updated section 3.3 to better explain the experimental setup for author identification.

We once again thank the reviewers for their helpful comments. We have responded to the individual issues raised by each of them. The revised version of VeriDark is more polished thanks to their feedback.

If there are any other questions during this discussion stage, we will gladly elaborate further on our responses.

---

### Meta-Review · Program_Chairs · 2022-09-16

**Recommendation:** Accept
**Confidence:** 3

**Metareview:**

This is likely an important dataset for the development of forensic tools for the dark web. At the same time, it presents a balancing act between making this dataset available and avoiding non-ethical use. The authors do go a long way in addressing these concerns, and most reviewers also feel that their other concerns are addressed. As a whole, I feel like this paper could be accepted.

Nevertheless, given the importance of proper use for this dataset, I would make acceptance conditional on two items:
* Include a proper datasheet, using for instance the format described here: https://arxiv.org/abs/1803.09010
Currently, the needed information on the dataset's use is fragmented between the paper, the supplementary, the github repo, and the access request. This needs to be brought together into a single document. The current supplementary is very limited.
* Document better how access to the dataset must be requested and what the procedure is. It is currently not very clear that one needs to use the Zenodo link. The alternative links to Google drive should probably be removed since a Google drive file is not an ideal archival storage medium. Also, it provides no trace of access requests made. In addition, it must be clear who can approve access and according to which rules. If only a single person can grant permission then this may raise concerns about the longer-term availability, and long-term availability is a key requirement for the NeurIPS datasets track.

---

### Decision · Program_Chairs · 2022-09-16

**Decision:**

Accept

**Comment:**

The meta-review contains important requirements before the paper can be accepted to the NeurIPS datasets and benchmarks track.
This includes the inclusion of better dataset documentation and assurances on long-term accessibility. Given the sensitive nature of the dataset, these issues need to be resolved before the paper can be finally accepted.